# Closed-Loop Supply Chain Network Equilibrium Model with Subsidy on Green Supply Chain Technology Investment

**Haixiang Wu [1,2], Bing Xu [1,*] and Ding Zhang [3]**

[1]   School of Management, Nanchang University, Nanchang 330031, China
[2]   Jiangxi Institute of Economic Administrators, Nanchang 330088, China
[3]   School of Business, State University of New York, Oswego, NY 13126, USA
*   Correspondence: xubing99@ncu.edu.cn

**Abstract:** The green supply chain (GSC) can effectively reduce the waste of resources and avoid environmental pollution. For a closed-loop supply chain network consisting of multiple manufacturers, multiple retailers, and multiple consumer and recycling markets, we assume that retailers are responsible for the recycling of used products, manufacturers use raw materials to produce new products and recycled products for remanufacturing, and government departments subsidize all manufacturers and retailers for GSC technology investment. Then, the equilibrium conditions of manufacturers, retailers, demand markets, and recycling markets are obtained by using the variational inequality method, complementarity theorem, and Nash equilibrium theory, and the variational inequality model of the closed-loop supply chain network multiphase equilibrium is established. Based on numerical simulation, the optimal technology investment decision of green supply chain under different government subsidy rates, and the influence of market structure and enterprise cost asymmetry on the equilibrium solution of supply chain network are analyzed. The results show that government subsidies can effectively promote enterprises to upgrade their level of GSC technology investment. The intensification of enterprise competition and the asymmetry of enterprise costs will affect the composition of enterprise profits and the allocation of profits between enterprises, and the former will weaken the effect of government subsidies.

**Keywords:** closed-loop supply chain network; GSC technological investment; government subsidy; equilibrium model; variational inequality

---

## 1. Introduction

The green supply chain (GSC), which aims to reduce resource waste and environmental pollution, has recently attracted extensive attention from both academia and industry. Green technology investment of the supply chain is defined as spending resources, in the form of money or knowledge, to acquire and deploy technologies that relate to waste recycling, energy saving, green product designs, pollution prevention, or environmental management [1]. The implementation of the GSC in core enterprises of the supply chain can enhance the environmental protection image of enterprises, save costs, and gain competitive advantages [2,3]. For this reason, many powerful large enterprises have put GSC management into practice and obtained fruitful returns. For example, Wal-Mart, the world's largest retailer, is committed to the use of clean energy and recyclable packaging materials, limiting products made from deforestation, and encouraging and supporting suppliers to reduce carbon emissions, which have generated considerable economic benefits for suppliers and themselves. In 2015, Siemens announced that it would spend USD 110 million to reduce its carbon emissions by 2020.

The tech company plans to cut half of its carbon emissions by 2020 and achieve carbon neutrality by 2030. Since fiscal 2014, they have reduced carbon emissions by nearly one-third, from 2.2 million tons to 1.5 million tons in fiscal 2018. 80% of the electricity they provided has been converted into renewable energy [4]. It expects that this investment will save the company between USD 20 million and USD 30 million annually [5]. Dell claims that packaging materials made from wheat straw save 40% of energy and 90% of water compared with traditional materials, and cost less [6]. Adidas, Coca-Cola, and other companies have raised GSC management to a strategic level of the company [7]. Since 2009, Shanghai General Motors has carried out the "excellent green supplier" selection and award activities. By 2015, nearly 300 suppliers had jointly invested in energy saving and emission reduction projects, which saved more than CNY 350 million in energy costs.

Large enterprises are actively involved in GSC technology investments, but this is not necessarily the case for small and medium-sized enterprises (SMEs). More than half of China's carbon emissions come from small and medium-sized enterprises [8]. Therefore, pollution emission control of small and medium-sized enterprises is also a serious problem that cannot be ignored. Because the implementation of the GSC will increase the cost of production and operation, enterprises are forced to increase the price of products due to short-term economic benefits, which will inevitably lose some price-sensitive customers, and thus reduce the short-term competitiveness of enterprises. The existence of price-sensitive customers and peer competition, coupled with the fact that GSC investment requires more capital investment and technology accumulation, as well as being ineffective in the short-term and involving high risk, amongst other characteristics, makes small and medium-sized enterprises reluctant to invest in GSC technology. In the long run, however, if SMEs want to seek development and growth in fiercely competitive markets without being eliminated, they must follow the example of large enterprises to invest in green technology, so as to comply with government environmental laws and meet the increasing green needs of consumers. In order to solve this dilemma and seek long-term stable and sustainable economic development, government subsidies for green technology investment are a policy option. As such, some local governments of China have formulated corresponding fiscal subsidy policies. Dongguan, the first pilot demonstration city of the GSC in China, put forward the Dongguan index of the GSC in 2016. It evaluates the five pilot industries of electronics, machinery, retail, furniture, and footwear every year, and gives subsidies of CNY 100,000 and CNY 150,000 to enterprises that have received four-star and five-star evaluations, respectively. Shanghai launched a GSC project plan in 2016, granting 30–50% financial subsidies to enterprises which have declared and been approved an energy-saving and emission reduction project. However, these measures are yet to be promoted nationwide.

Technology investment and government subsidies have long-term effects. GSC investment may increase the cost and accounting risk of enterprises in the short-term but, in the long-term, it can save costs, increase profits, and enhance the competitiveness of enterprises due to the expansion of demand for green products driven by the improvement of consumers' environmental awareness. Government subsidies may not be effective in the short-term, but in the long-term, they may mobilize the enthusiasm of enterprises and affect the allocation of resources. The correct depiction of the long-term effects of government subsidies for technology investment in the GSC is an issue worth discussing and is the focus of this paper. In addition, market structure also has an important impact on government subsidies. Does the increase of market members make enterprises more willing to expand their investment scale to obtain more subsidies, or is investment weakened by the scarcity of resources and the limitation of demand? This is also one of the problems studied in this paper. Moreover, in a supply chain with many members, the strength of these enterprises is often different. This paper also studies the influence of government subsidies on the behavior and interests of different enterprises when the strength of enterprises is asymmetric.

This paper attempted to resolve the dilemma encountered by SMEs in GSC technology investment, studied from the perspective of government subsidies, in order to ultimately achieve long-term stable and sustainable development of the economic, social, and ecological environment.

The remainder of this paper is organized as follows. We review related literature in Section 2. In Section 3, the notations and assumptions of the model are described. In Section 4, we use variational inequalities to derive the equilibrium conditions of manufacturers, retailers, and demand markets, and finally obtain the conditions for the equilibrium of the network. Three numerical examples are presented in Section 5, which aim to verify the validity of the model and analyze the effects of increased competition and asymmetric membership costs on the equilibrium results. In Section 6, we conduct a series of discussions from the perspectives of the economic development history of developing countries, international trade, and existing subsidies in developed countries, and put forward some insights on management and economics. We summarize our paper in Section 7.

## 2. Literature Review

This work is closely related to green technology investment of the supply chain, government subsidies based on technology investment of the GSC, and supply chain network equilibrium.

### 2.1. Green Technsology Investment of Supply Chain

Many scholars have studied the green technology investment of enterprises. These studies provide some references for the study of this paper. Doval and Negulescu investigated the drivers of Romanian enterprises to implement green investment in the form of questionnaires and found that the top three drivers are the intensification of competition among enterprises, the shortage of resources, and the pressure of government environmental protection law [9]. Gurnani, Erkoc, and Luo studied the price and investment decisions of a supply chain consisting of a manufacturer with product quality investment and a retailer with sales effort investment under different decision orders, such as product quality coefficient, sales effort coefficient, wholesale price, and retail price [10]. Yang, Xiao, and Huang studied a channel selection problem between the traditional retail channel and online direct sell channel for an environmentally responsible manufacturer with GSC investment, and concluded that when the environmental cost of retail channel is high, manufacturers will tend to open an online direct channel [11].

Stucki found that only 19% of the highest energy cost firms have positive marginal productivity of GSC technology investment [12]. Ma, Zhang, Hong, et al. considered a two-stage supply chain composed of two manufacturers producing green products and ordinary products, respectively, and one retailer, and studied the price and green degree decision-making models under six modes, including different decision-making sequences and green investment cost sharing [13]. Zhang, Liu, Zhang, et al. studied investment in the preservation technology of perishable goods under the monopoly of a manufacturer and a retailer, and proposed a technology investment cost–revenue sharing contract to coordinate the benefits of all parties [14].

Jin, Zhang, Liu, et al. used economics and econometrics to analyze 17 years of panel data of 30 provincial administrative regions in China, and concluded that public appeal can promote enterprises to increase green investment and local governments to implement stricter environmental regulation [15].

Yan, Shi, Ye, et al. assumed that the use of radio-frequency identification (RFID) techniques can reduce losses in the transportation of fresh agricultural products. On this premise, the application of a radio frequency technique by manufacturers was studied, and a revenue-sharing contract between manufacturer and retailer was proposed to promote the application of this technique [16]. Yang, Miao, and Zhao studied trade credit issues between a manufacturer with GSC technology investment and two retailers [17]. Cai, Lai, and Liu et al. proposed a state space model to verify the effectiveness of lean energy saving and an emission reduction strategy in promoting the sustainability of a manufacturing industry [18].

In most of the above literatures, the Stackelberg game models with no more than three members were established, and the backward induction method was used to solve the model. Some used empirical methods to study the reality of green technology investment in the supply chain. Unlike these,

this paper intends to study the technology investment subsidy problem under a supply chain network with more participants' intense competition.

## 2.2. Government Subsidies Based on Technology Investment of the Green Supply Chain

In recent years, many scholars have undertaken significant research into green supply chains under government subsidies. The main methods adopted are the Stackelberg game model, the empirical method, and the evolutionary game method.

Mitra and Webster considered the two-stage price decision model of single or simultaneous government subsidies to producers and remanufacturers, and concluded that appropriate government subsidies to producers would help producers design products that are easier to recycle [19]. Madani and Rasti-Barzoki studied a Stackelberg game model of government price subsidies for green production and taxes for non-green production and considered government revenue from the perspective of ecological restoration, the government tax, and the subsidy. It was concluded that a subsidy could generate more government and enterprise revenue, as well as better promote the development of green products, than taxation [20]. Yi and Li studied supply chain coordination with the coexistence of subsidies for energy-saving investments and taxes on carbon emissions. Manufacturers have to trade-off investments between energy saving and emission reduction [21]. Wan and Hong established a Stackelberg game model under two subsidy modes of government subsidies for remanufacturing and recycling, respectively. The structure involved is a closed-loop supply chain composed of a manufacturer, a retailer, and a recycler [22]. Chen, Dimitrov, and Pun studied a tripartite Stackelberg game model in which manufacturer and retailer jointly invest in product sustainability and therefore enjoy government subsidies. The model takes into account the environmental, economic, and social benefits of the government [23].

Liu et al. proposed a Stackelberg game model consisting of government, multiple suppliers, and a socially responsible retailer. The government can only subsidize the retailer for their social responsibility. The government targets include consumer surplus, corporate profits, and subsidized expenditure. The optimal subsidy coefficient of the government is obtained by using the backward induction method [24]. Gao et al. also demonstrated the importance of government green standards and environmental subsidies to manufacturers by establishing a Stackelberg game model [25]. Giri et al. established a Stackelberg game model under different decision-making orders of government, two manufacturers, and one retailer. The model also considered the competition and cooperation among members [26].

Sun et al. established an evolutionary game model of green investment decision-making of supplier and manufacturer groups under the background that both supplier and manufacturer's green investment can be subsidized by government. It is concluded that setting government subsidies in a certain range can reduce the free-riding behavior of green investment in the two groups [27]. Liu et al. analyzed the game behavior of government and manufacturer by establishing an evolutionary game model [28]. On this basis, Chen et al. demonstrated that the dynamic subsidy mechanism is more effective than other tax and subsidy mechanisms with static incentives for manufacturers to adopt low-carbon manufacturing [29].

Bai et al. used empirical methods to prove that government RD subsidies for energy-intensive enterprises can effectively promote their green innovation and improve their trends and performance [30]. Bai, Hua et al. demonstrated the positive effect of government environmental subsidies on the green efficiency of thermal power enterprises by empirical methods [31]. Using real data analysis, Yang et al. demonstrated that government subsidies are the main force to support small and medium-sized renewable energy enterprises to carry out green innovation [32]. Nicolini et al. validated the effective role of energy subsidies in promoting renewable energy in Europe through data analysis [33].

By establishing an optimization model, Huang et al. demonstrated that the interest rate of green credit provided by the government must be less than a certain threshold and the loan scale must be greater than a certain threshold, so that enterprises can carry out effective green innovation [34].

Xiao et al. studied the supply chain consisting of an ordinary product manufacturer and a green product manufacturer. Under the background that the government only subsidizes the green product manufacturer, considering the multi-objective decision-making problems of government revenue, employment rate, and carbon emissions, the optimal government subsidy rate was obtained by using a data envelopment analysis technique [35].

The Stackelberg game model can effectively describe the members' sequential decision-making process. In the problem of government subsidies, it can also achieve the government's multi-criteria objectives and obtain the formula solution of the model, and conveniently uses derivatives to analyze its monotony, concave, and convex qualitative properties. The evolutionary game model can effectively describe the game behavior and evolution direction of two groups. However, the model has fewer players and the decision variables can only take a few discrete values. For example, in the above model, the supplier and the remanufacturer decide whether to participate in green investment. In this case, the number of decision-making bodies is two, and the decision variables can take two values [28]. The empirical method can make good use of real data to verify the correlation between two variables, but it cannot describe the game relationship between members. Differently from the above methods, variational inequalities have advantages in dealing with game equilibrium problems with a large number of decision makers and decision variables. For example, in the model of this paper, there are seven decision makers and 167 decision variables in a supply chain network composed of three manufacturers, two retailers, and two demand markets when the planning period is five.

Similar to this paper is Reference [36], which used variational inequality as a tool to obtain the equilibrium output of manufacturers' products with different green grades based on the background of government subsidies for new-energy vehicles and the premise that the government provides different subsidy rates for manufacturers' products with different green grades. Differently from this, the present paper considers multi-layer, multi-period, and green degree as continuous value.

*2.3. Supply Chain Network Equilibrium*

The supply chain network equilibrium model is an effective tool for describing competition and cooperation among members. Nagurney used variational inequalities and Nash equilibrium theory to analyze the optimality conditions of members at all levels of the supply chain network, and creatively proposes a super supply chain network equilibrium model [37]. Later, many scholars extended the model to encompass multiple channels [38], multiple criteria [39–41], demand type [42], demand risk [43], and multiple periods [44]. In addition, some research has studied the issues of government levying carbon emission taxes [45] and pollution emission permit trading [46].

Saberi et al. studied the supply chain network equilibrium model of technology investment in GSC in the initial stage of enterprises, and concluded that the investment can save the costs of enterprises and increase the overall profits of the supply chain network in the long run [5]. Based on this, our paper further considers the effect of government subsidies on enterprises' GSC technology investment and the recycling and remanufacturing of waste products. The contribution of this paper has three aspects: First, a multi-phase supply chain network equilibrium model was built to analyze the long-term impact of government subsidies and the long-term decision-making of enterprises. Second, competition involving multi-tier members was considered, which is more common in reality. Finally, the effect of government subsidies in a multi-party competitive situation was analyzed by establishing mathematical models. This is the first time a multiphase closed-loop supply chain network equilibrium model of government-subsidized GSC technology investment has been constructed to analyze the efficiency of government subsidies, which can effectively promote the enthusiasm of small and medium-sized enterprises to invest in GSC technology.

## 3. Problem Statement and Formulation

### 3.1. Problem Description

Consider a three-tier closed-loop supply chain network: the first tier is composed of *M* manufacturers producing products through raw materials and recycled materials; the second tier comprises *N* retailers responsible for product sales and waste product recycling; and the third tier comprises *K* demand and recycling markets. Its network structure is shown in Figure 1, where nodes represent network entities, real connection lines represent positive flows of production and sales, and virtual connection lines represent reverse flows of waste product recycling.

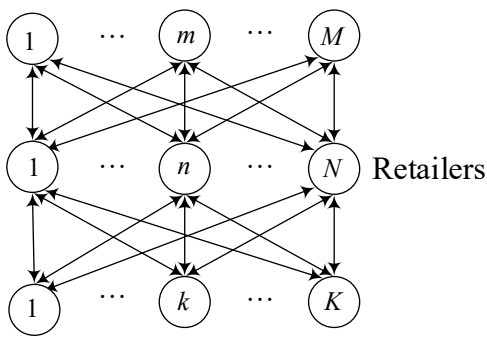

**Figure 1.** Closed-loop supply chain network structure.

This paper studies a multiphase closed-loop supply chain network equilibrium model. Suppose that in the first phase, the manufacturer uses raw materials to make a product at a certain rate, then sells it to the retailer, and the retailer sells it to the consumer in the demand market. At the end of the first phase (equivalent to the beginning of the second phase), retailers recycle waste products in the demand market which will then be disassembled, selected, and converted into reusable recycling materials at a certain ratio. After that, manufacturers buy reusable materials from retailers and convert them into remanufactured products at a certain rate. That is to say, new products and remanufactured products will be sold on the market at the same time from the second phase. In the final phase, retailers no longer recycle used products.

### 3.2. Model Assumptions

(1) New products and remanufactured products made by different manufacturers are homogeneous.

(2) The members of the same network layer compete against each other, the members of the heterogeneous network cooperate with each other, and the information among them is completely symmetrical.

(3) The effect of technology innovation made by manufacturers and retailers is expressed as the green rate (GR), whose process consists of production, transportation, warehousing, sales, etc., which aims to save energy and resources, and reduce environmental hazards. The larger the value, the more energy and resources are saved and the less hazardous substances are discharged in the above process. Manufacturers and retailers must spend money to invest in GSC technology innovation in order to achieve a certain degree of greenness.

(4) Retailers (such as Wal-Mart) have environmental preferences and choose to trade only with manufacturers whose greenness is no less than their own, which means the greenness of the manufacturers with which they trade must not be less than that of the retailer.

(5) Government subsidizes manufacturers and retailers for their green technology investment. The calculation formula of the subsidy is expressed as

$$Subsidy\ amount = \ Subsidy\ coefficient \times Green\ degree \times Product\ quantity$$

(6) Waste generated by manufacturers in the manufacturing and remanufacturing process and waste generated by retailers in the selection process of recycled products will be transported to landfills for burial. Manufacturers and retailers must pay for landfill treatment fees.

(7) In the early stages of the supply chain, manufacturers and retailers decide the level of GSC investment in order to reach some certain green degree.

### 3.3. Notations and Technical Scheme

The symbols and explanations of variables and functions are shown in Table 1. In Figure 2, the main variables are divided into three types. In summary, the transaction volume of all the items between the upper and lower layers are decision variables. Except for the prices that the consumer is willing to pay for products, which are decision variables, all other prices are endogenous variables. Inventory quantities and green rates are decision variables, and government subsidization rates are exogenous variables. The symbol " ⬍ " means that when the model reaches equilibrium, the volumes and prices related to transactions between upper and lower layers must be coincident, the amount of material shipped out must be equal to the amount shipped in, and the price which is willingly paid must be equal to that willingly accepted.

Figure 3 is the technical framework of this paper. Firstly, we assume that each manufacturer is rational, which means that each manufacturer chooses the most appropriate strategy to achieve the goal of optimal profit under the given decision of other manufacturers. In this context, the decision results of each manufacturer satisfy the Nash equilibrium, which is equivalent to a variational inequality. Similar to the manufacturer level, the equilibrium results of all parties in the retailer level can also be expressed by another variational inequality. The decision-making of consumer and recycling markets are spatial price equilibrium problems, that is, complementarity problems, which are equivalent to the variational inequality problem, thus the third variational inequality is obtained. As mentioned above, when the network reaches equilibrium, the amount of goods that upper and lower members are willing to ship-out must equal the amount that they are willing to accept. This means we can add these three variational inequalities together to obtain a global variational inequality.

The problem of network equilibrium is formulated as solving a variational inequality. We introduce a projective contraction algorithm for solving this variational inequality. Then three different examples are presented. Combining algorithm, global variational inequality, and these examples, three corresponding computer programs are compiled. The results of three examples are obtained by running these programs with MATLAB software (version 7.1.0.246(R14)Service Pack 3, The Mathworks, Inc., Natick, MA, USA). After, we analyzed the results. At the end of the article, we summarize the paper and put forward some enlightenment of management and economics.

## Table 1. Variables, parameters, and functions.

| Notations | Descriptions |
|---|---|
| $t, m, n, k$ | A specific period, a specific manufacturer, a specific retailer and a specific demand market, where: $t \in \{1, \ldots, T\}$, $m \in \{1, \ldots, M\}$, $n \in \{1, \ldots, N\}$, $k \in \{1, \ldots, K\}$. |
| $\gamma$ | Supply chain net present value (NPV) discount rate. |
| $\beta_r, \beta_u, \chi$ | The conversion rate of raw materials to products (manufacture conversion rate), the conversion rate of waste materials to new products (remanufacture conversion rate), and the ratio of rec products to reusable materials at retailers (reusable conversion rate), |
| $\delta_m, \delta_n, \underline{\delta}_n$ | The green rate of manufacturer m and retailer n respectively, and the minimum green degree that retailer must achieve in order to meet the government's minimum environmental requirements. The higher the value of greenness, the more energy and resources are saved, and the less harmful to the environment. The first two constitute $M$ dimension column vectors $\delta^M$ and $N$ dimension column vectors $\delta^N$ respectively. |
| $\omega_m, \omega_n$ | The subsidization rates of the government to manufacturer and retailer per unit green degree product. |
| $\overline{\rho}$ | Unit waste product or waste material treatment fee. |
| $l_{tk}$ | The highest recovery rate of waste products in the consumer market k of the period t. |
| $q_{tm}^r$ | Material utilization of manufacturer m in phase t. Such quantity of manufacturer m in each period constitutes $T$ dimension column vector $q_m^r$, these category quantity of each manufacturer and each phase constitutes a $T \times M$ dimension column vector $q^r$, and these category quantity of phase t constitutes $M$ dimension column vector $q_t^r$. |
| $q_{tm}^u$ | Recycled material utilization of manufacturer m phase t. Such quantity of manufacturer m in each period constitutes a $T-1$ dimension column vector $q_m^u$, the category quantity of each manufacturer and each phase constitutes a $(T-1) \times M$ dimension column vector $q^u$, and the category quantity of phase t constitutes an $M$ dimension column vector $q_t^u$. |
| $q_{tmn}^1, p_{tmn}^1$ | The volume and price of product transactions between manufacturer m and retailer n in period t, respectively. Such transaction quantity of manufacturer m in each period constitutes a $T \times N$ dimension column vector $q_m^1$. Such transaction quantity of retailer n in each period constitutes a $T \times M$ dimension column vector $q_n^1$. The category volume and price of transaction products between every pair of manufacturers and retailers in every period constitutes a $T \times M \times N$ dimension column vector $Q^1$ and $P^1$, respectively. The category volume and price of phase t also constitutes an $M \times N$ dimension column vector $Q_t^1$ and $P_t^1$, respectively. |
| $q_{tnk}^2, p_{tnk}^2$ | The volume and price of product transactions between retailer n and consumers in demand market k in the period t, respectively. Such transaction quantity of retailer $n$ in each period constitutes a $T \times K$ dimension column vector $q_n^2$, the category volume and price of transaction products between every pair of retailers and markets in every period constitutes a $T \times N \times K$ dimension column vector $Q^2$ and $P^2$, respectively, and the category volume and price of phase t constitutes an $N \times K$ dimension column vector $Q_t^2$ and $P_t^2$, respectively. |
| $q_{tnk}^3, p_{tnk}^3$ | The volume and price of recycling product transactions between retailer $n$ and consumers in market $k$ in the period $t$, respectively. Such transaction quantity of retailer $n$ in each period constitutes a $(T-1) \times K$ dimension column vector $q_n^3$, the category volume and price of transaction products between every pair of retailers and markets in every period constitutes a $(T-1) \times N \times K$ dimension column vector $Q^3$ and $P^3$, respectively, and the category volume and price of phase t constitutes an $N \times K$ dimension column vector $Q_t^3$ and $P_t^3$, respectively. |
| $q_{tmn}^4, p_{tmn}^4$ | The volume and price of recycling material transactions between manufacturer m and retailer n in period t, respectively. Such transaction quantity of manufacturer m in each period constitutes a $(T-1) \times N$ dimension column vector $q_m^4$. Such transaction quantity of retailer n in each period constitutes a $(T-1) \times M$ dimension column vector $q_n^4$, and the category volume and price of transaction products between every pair of manufacturers and retailers in every period constitutes a $(T-1) \times M \times N$ dimension column vector $Q^4$ and $P^4$, respectively. The category volume and price of period t also constitutes an $M \times N$ dimension column vector $Q_t^4$ and $P_t^4$, respectively. |
| $p_{tnk}^5, D_{tnk}(p_t^5, \delta_n)$ | The prices of products that consumers in market $k$ are willing to pay and the corresponding demand amount sold by retailers $n$. The price that consumers in all consumer markets are willing to pay in period $t$ constitutes an $N \times K$ dimension vector $p_t^5$, and the category prices related to every retailer in every period constitutes a $T \times N \times K$ dimension column vector $P^5$. $D_{tnk}(p_t^5, \delta_n)$ is assumed to be a monotonic decreasing function of $p_{tnk}^5$. In order to reflect consumers' preference for green products, we assume that $D_{tnk}(p_t^5, \delta_n)$ is a monotonic increasing function of green degree $\delta_n$. |
| $I_{tm}^M, I_{tn}^N$ | Inventory quantities of manufacturer m and retailer n in period t. The inventory of every manufacturer and every retailer in each period constitutes a $T \times M$ dimension column vector $I^M$ and $T \times N$ dimension column vector $I^N$, respectively. The category inventory of period t constitutes an $M$ dimension column vector $I_t^M$ and $N$ dimension column vector $I_t^N$, respectively. |
| $f_{tm}^r(q_t^r, \beta_r, \delta_m)$ | The manufacturing cost function of the manufacturer m in period t, which is the continuous differentiable convex function of $q_{tm}^r$. It also depends on manufacture conversion rate $\beta_r$ and manufacturer m's green degree $\delta_m$. |

**Table 1.** *Cont.*

| Notations | Descriptions |
|---|---|
| $f_{tm}^u(q_t^u, \beta_u, \delta_m)$ | The remanufacturing cost function of manufacturer m from using raw materials in period t, which is assumed to be the continuous differentiable convex function of $q_{tm}^u$. It also depends on remanufacture conversion rate $\beta_u$ and manufacturer m's green degree $\delta_m$. |
| $C_{tmn}^{rM}(Q_t^1, \delta_m), C_{tmn}^{rN}(Q_t^1, \delta_n)$ | The product transaction costs assumed by manufacturer m and retailer n, respectively, which are associated with the product transactions between them, and are assumed as the continuous differentiable convex functions of $q_{tmn}^1$. |
| $C_{tmn}^{uM}(Q_t^4, \delta_m), C_{tmn}^{uN}(Q_t^4, \delta_n)$ | The recycling material transaction costs assumed by manufacturer m and retailer n, respectively, which are associated with the recycling material transactions between them, and are assumed as the continuous differentiable convex functions of $q_{tmn}^u$. |
| $C_{tnk}^{rN}(Q_t^2, \delta_n), C_{tnk}(Q_t^2, \delta_n)$ | The product transaction costs assumed by retailer *n* and consumers in market *k*, respectively, which are associated with the product transactions between them, and are assumed as the continuous differentiable convex functions of $q_{tnk}^u$. |
| $C_{tnk}^{uN}(Q_t^3, \delta_n)$ | The recycling product transaction costs assumed by retailer n, which are associated with the recycling product transaction between retailer n and consumers in market *k*, and are assumed as the continuous differentiable convex functions of $q_{tnk}^u$. |
| $C_{tn}(Q_t^2, \delta_n)$ | Retailer *n*'s product exhibition and advertising expenses in period *t*. In order to reflect competition, let it depend on $Q_t^2$, the entire product transaction vector between all retailers and consumers in period *t*, and which is the continuous differentiable convex function of $q_{tn}^2$, the product ship between retailer n and consumers on each market *k* in period *t*. |
| $C_{tn}^u(Q_t^3, \delta_n)$ | The cost of disassembling, cleaning and picking of recycled products assumed by retailer *n* in period *t*. In order to reflect competition, suppose it depends on $Q^3$, the entire transaction volume vector on this layer, and is the continuous differentiable convex function of $q_{tnk}^3$. |
| $V_m(\delta^M), V_n(\delta^N)$ | The GSC technology investment of manufacturer *m* and retailer *n*, assumed as the continuous differentiable convex function of $\delta_m$ and $\delta_n$, respectively. |
| $H_{tm}(I_t^M, \delta_m), H_{tn}(I_t^N, \delta_n)$ | The inventory cost of manufacturer *m* and retailer *n*, assumed as the continuous differentiable convex function of $I_{tm}^M$ and $I_{tn}^N$, respectively. |
| $\alpha_{tnk}(Q_t^3, \delta_n)$ | The negative utility of consumers in market *k* when returning used products in period *t*, reflecting the consumers' feeling of aversion at the above process, is a monotonic increasing function of $q_{tnk}^u$. In order to reflect competition, it is assumed to depend on $Q_t^3$, a vector grouped by all the recycling products between retailers and consumers in period *t*. |

Notes: In order to reflect competition on resources, it is assumed all cost functions, such as $f_{tm}^r(q_t^r, \beta_r, \delta_m)$, $f_{tm}^u(q_t^u, \beta_u, \delta_m)$, $C_{tmn}^{rM}(Q_t^1, \delta_m)$, and $C_{tmn}^{rN}(Q_t^1, \delta_n)$ depend on all the quantities possessed by members on the same layer or the shipped quantities between the same layer pairs. In the long run, improving the greenness of the supply chain can save costs, so cost functions such as $f_{tm}^r(q_t^r, \delta_m)$, $f_{tm}^u(q_t^u, \beta_u, \delta_m)$, $C_{tmn}^{rM}(Q_t^1, \delta_m)$, and $C_{tmn}^{rN}(Q_t^1, \delta_n)$ are monotonous reduction functions of $\delta_m$ or $\delta_n$.

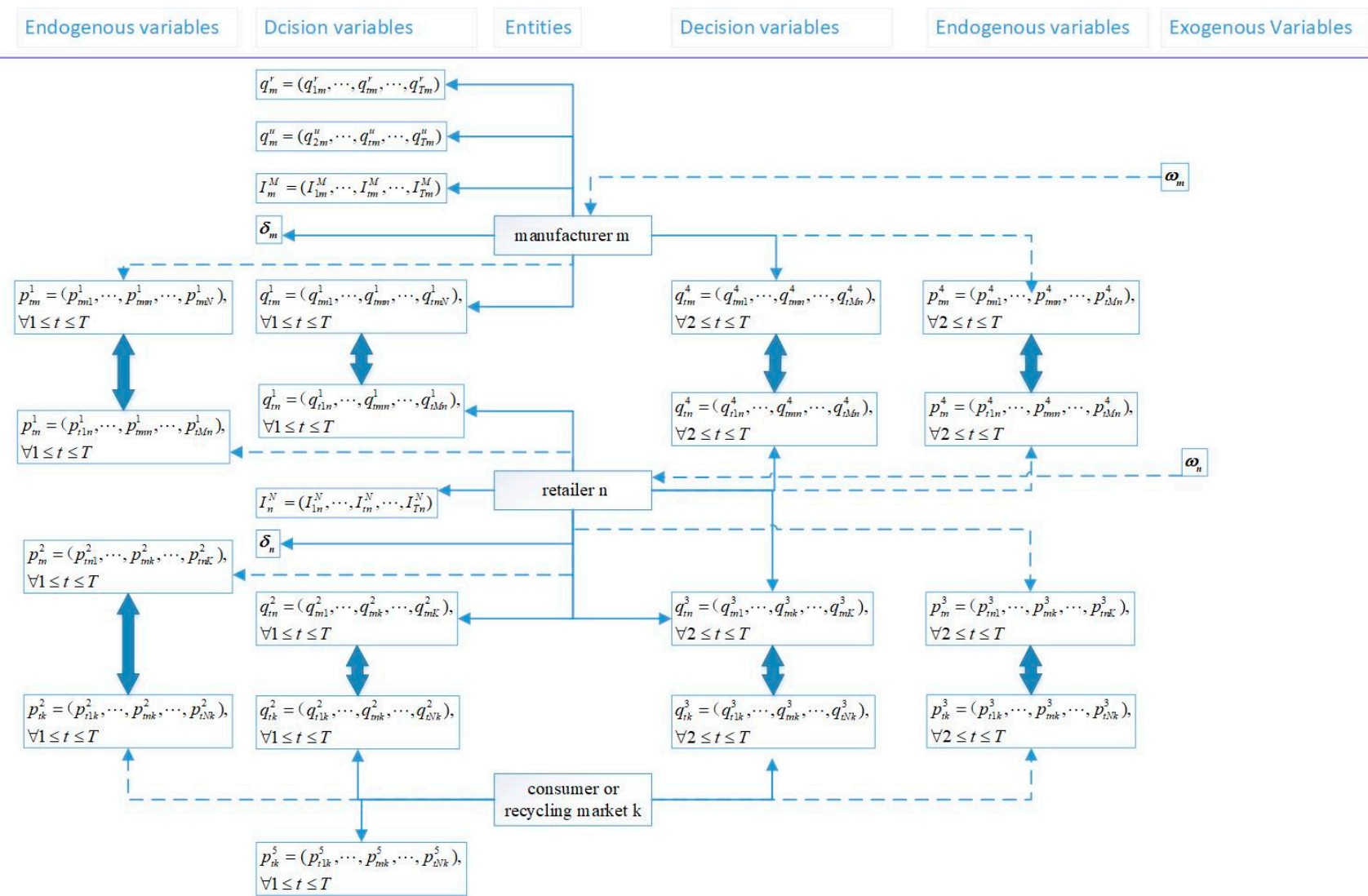

**Figure 2.** Types of variables.

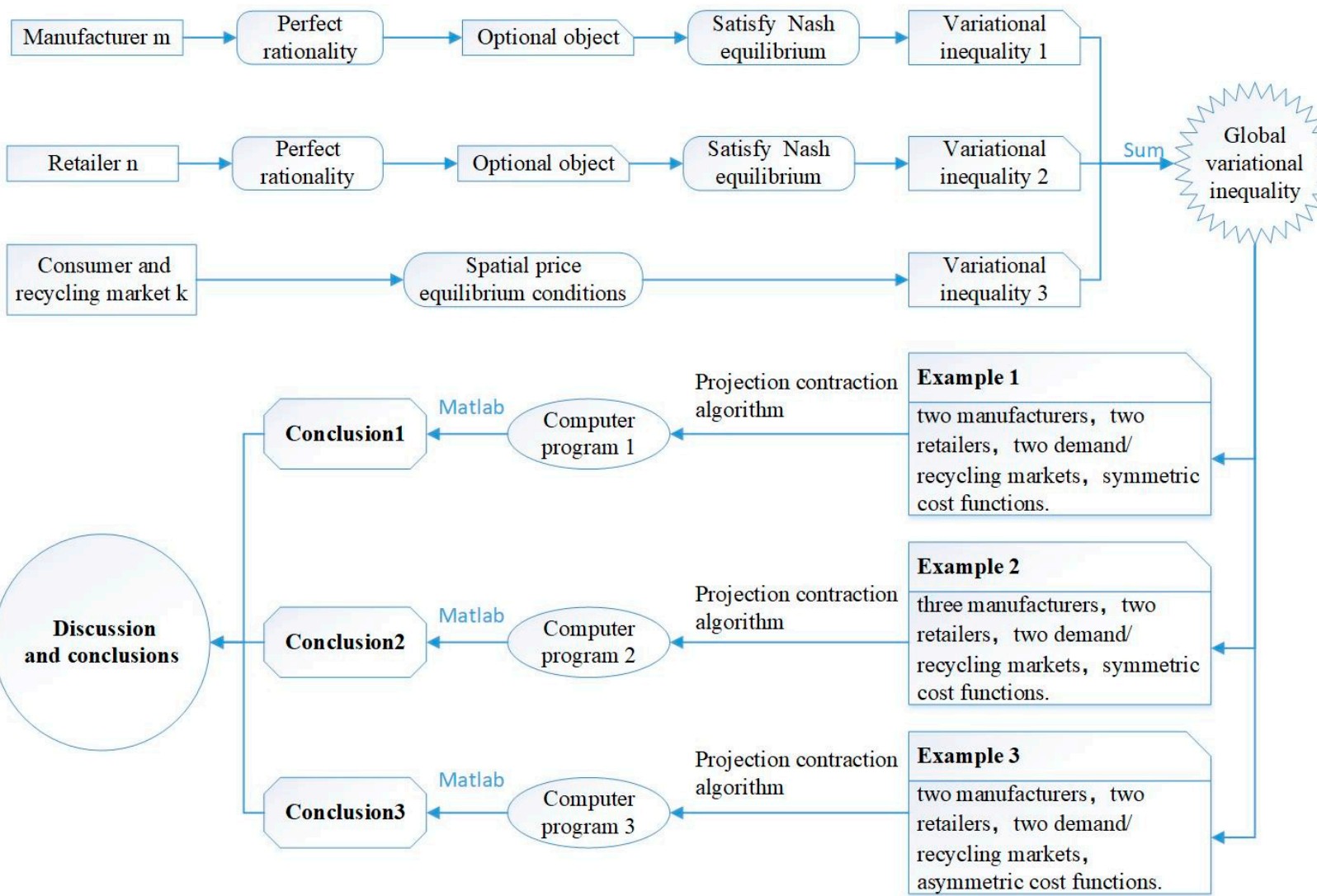

**Figure 3.** Technical scheme.

## 4. Variational Inequality Equilibrium Model for Multi-Period Closed-loop Supply Chain Networks

### 4.1. Behavior of the Manufacturers and Their Equilibrium Conditions

The decision of manufacturer *m* will involve deciding the quantity of raw and recycling materials used to make a product, the quantity of the product and the used material traded with every retailer, the inventory in every period, and the green degree of its supply chain, which aims to maximize its profit. Its decision model can be expressed as

$$
\max \pi_m(q_m^r, q_m^u, q_m^1, q_m^4, I_m^M, \delta_m) = \sum_{t=1}^{T} \frac{1}{(1+\gamma)^t} \left\{ \sum_{n=1}^{N} p_{tmn}^1 q_{tmn}^1 - f_{tm}^r(q_t^r, \beta_r, \delta_m) - \sum_{n=1}^{N} C_{tmn}^{rM}(Q_t^1, \delta_m) - H_{tm}(I_t^M, \delta_m) + \omega_m \delta_m \sum_{n=1}^{N} q_{tmn}^1 \right\}
$$
$$
- \sum_{t=2}^{T} \frac{1}{(1+\gamma)^t} \left\{ f_{tm}^u(q_t^u, \beta_u, \delta_m) + \sum_{n=1}^{N} p_{tmn}^4 q_{tmn}^4 + \sum_{n=1}^{N} C_{tmn}^{uM}(Q_t^4, \delta_m) + \overline{\rho}(1-\beta_u) \sum_{n=1}^{N} q_{tmn}^4 \right\} - V_m(\delta^M) \tag{1}
$$

s.t:

$$
\beta_r q_{tm}^r \geq I_{tm}^M + \sum_{n=1}^{N} q_{tmn}^1, t = 1 \tag{2}
$$

$$
I_{(t-1)m}^M + \beta_r q_{tm}^r + \beta_u q_{tm}^u \geq I_{tm}^M + \sum_{n=1}^{N} q_{tmn}^1, t \in \{2, \cdots, T\} \tag{3}
$$

$$
\sum_{n=1}^{N} q_{tmn}^4 \geq q_{tm}^u, t \in \{2, \cdots, T\} \tag{4}
$$

where $q_{tm}^r, q_{tm}^u, q_{tmn}^1, q_{tmn}^4, I_{tm}^M, \delta_m \geq 0, \forall t \in \{1, \cdots, T\}, m \in \{1, \cdots, M\}, n \in \{1, \cdots, N\}$.

The profit of manufacturer m in period *t* includes the sum of the sales revenue $\sum_{n=1}^{N} p_{tmn}^1 q_{tmn}^1$ and the subsidy income $\omega_m \delta_m \sum_{n=1}^{N} q_{tmn}^1$ minus various costs, which include: manufacturing costs $f_{tm}^r(q_t^r, \beta_r, \delta_m)$; remanufacturing costs $f_{tm}^u(q_t^u, \beta_u, \delta_m)$; product and recycling material transaction costs $\sum_{n=1}^{N} C_{tmn}^{rM}(Q_t^1, \delta_m)$ and $\sum_{n=1}^{N} C_{tmn}^{uM}(Q_t^4, \delta_m)$, respectively; purchasing costs of used materials $\sum_{n=1}^{N} p_{tmn}^4 q_{tmn}^4$; inventory costs $H_{tm}(I_t^M, \delta_m)$; and disposal fees for unused waste materials $\overline{\rho}(1-\beta_u) \sum_{n=1}^{N} q_{tmn}^4$. Considering the time value of funds, the incomes and costs in different periods is discount to net present value (NPV), so every part above should be multiplied by the coefficient $\frac{1}{(1+\gamma)^t}$. In the early stages of the supply chain, manufacturers decide the level of GSC investment to obtain some green degree. Therefore, to derive the gross profits of the manufacturer *m*, the initial GSC investment must be subtracted from the sum of discounted profits of every period.

Equations (2)–(4) are flow conservation constraints. Equation (2) specifies that the sum of product volume trading with retailers and the inventory volume should not exceed the production quantity in the first phase. Equation (3) states that the sum of the product volume traded with retailers and the current inventory should not exceed the sum of products from using raw materials and recycling materials, and the previous inventory. Equation (4) specifies that the amount of waste material used by manufacturer *m* for remanufacturing shall not exceed the amount of waste material collected.

We assume that all manufacturers are perfectly rational and compete in a non-cooperative manner. According to the (Cournot–)Nash equilibrium conditions of the oligopoly state, each manufacturer will determine his own optimal collection, production and shipment by considering other manufacturers' optimal strategies. Therefore, the optimality conditions of all manufacturers can

be formulated simultaneously as the following inequality (see also Reference [37]): Determine $(q^{r*}, q^{u*}, Q^{1*}, Q^{4*}, I^{M*}, \delta^{M*}, \mu^*, \theta^*) \in R_+^{TM+(T-1)M+TMN+(T-1)MN+TM+M+TM+(T-1)M}$ satisfying

$$
\begin{aligned}
& \sum_{t=1}^{T} \sum_{m=1}^{M} \left\{ \frac{1}{(1+\gamma)^t} \partial f_{tm}^r(q_t^{r*}, \beta_r, \delta_m^*) / \partial q_{tm}^r - \beta_r \mu_{tm}^* \right\} \times [q_{tm}^r - q_{tm}^{r*}] \\
& + \sum_{t=2}^{T} \sum_{m=1}^{M} \left[ \frac{1}{(1+\gamma)^t} \partial f_{tm}^u(q_t^{u*}, \beta_u, \delta_m^*) / \partial q_{tm}^u - \beta_u \mu_{tm}^* + \theta_{tm}^* \right] \times [q_{tm}^u - q_{tm}^{u*}] \\
& + \sum_{t=1}^{T} \sum_{m=1}^{M} \sum_{n=1}^{N} \left\{ \frac{1}{(1+\gamma)^t} \left[ \partial C_{tmn}^{rM}(Q_t^{1*}, \delta_m^*) / \partial q_{tmn}^1 - p_{tmn}^{1*} - \omega_m \delta_m^* \right] + \mu_{tm}^* \right\} \times [q_{tmn}^1 - q_{tmn}^{1*}] \\
& + \sum_{t=2}^{T} \sum_{m=1}^{M} \sum_{n=1}^{N} \left\{ \frac{1}{(1+\gamma)^t} \left[ \partial C_{tmn}^{uM}(Q_t^{4*}, \delta_m^*) / \partial q_{tmn}^4 + p_{tmn}^{4*} + \overline{\rho}(1-\beta_u) \right] - \theta_{tm}^* \right\} \times [q_{tmn}^4 - q_{tmn}^{4*}] \\
& + \sum_{t=1}^{T-1} \sum_{m=1}^{M} \left\{ \frac{1}{(1+\gamma)^t} \partial H_{tm}(I_t^{M*}, \delta_m^*) / \partial I_{tm}^M + \mu_{tm}^* - \mu_{(t+1)m}^* \right\} \times [I_{tm}^M - I_{tm}^{M*}] \\
& + \sum_{m=1}^{M} \left\{ \frac{1}{(1+\gamma)^T} \partial H_{Tm}(I_T^{M*}, \delta_m^*) / \partial I_{Tm}^M + \mu_{Tm}^* \right\} \times [I_{Tm}^M - I_{Tm}^{M*}] \\
& + \sum_{m=1}^{M} \left\{ \begin{array}{l} \sum_{t=1}^{T} \frac{1}{(1+\gamma)^t} \left[ \partial f_{tm}^r(q_t^{r*}, \beta_r, \delta_m^*) / \partial \delta_m + \partial H_{tm}(I_t^{M*}, \delta_m^*) / \partial \delta_m + \sum_{n=1}^{N} \partial C_{tmn}^{rM}(Q_t^{1*}, \delta_m^*) / \partial \delta_m - \omega_m \sum_{n=1}^{N} q_{tmn}^{1*} \right] \\ + \sum_{t=2}^{T} \frac{1}{(1+\gamma)^t} \left[ \partial f_{tm}^u(q_t^{u*}, \beta_u, \delta_m^*) / \partial \delta_m + \sum_{n=1}^{N} \partial C_{tmn}^{uM}(Q_t^{4*}, \delta_m^*) / \partial \delta_m \right] + \partial V_m(\delta^{M*}) / \partial \delta_m \end{array} \right\} \times [\delta_m - \delta_m^*] \\
& + \sum_{m=1}^{M} [\beta_r q_{1m}^{r*} - I_{1m}^{M*} - \sum_{n=1}^{N} q_{1mn}^{1*}][\mu_{1m} - \mu_{1m}^*] + \sum_{t=2}^{T} \sum_{m=1}^{M} [I_{(t-1)m}^{M*} + \beta_r q_{tm}^{r*} + \beta_u q_{tm}^{u*} - I_{tm}^{M*} - \sum_{n=1}^{N} q_{tmn}^{1*}][\mu_{tm} - \mu_{tm}^*] \\
& + \sum_{t=2}^{T} \sum_{m=1}^{M} [\sum_{n=1}^{N} q_{tmn}^{4*} - q_{tm}^{u*}][\theta_{tm} - \theta_{tm}^*] \geq 0 , \forall (q^r, q^u, Q^1, Q^4, I^M, \delta^M, \mu, \theta) \in R_+^{TM+(T-1)M+TMN+(T-1)MN+TM+M+TM+(T-1)M}
\end{aligned}
\tag{5}
$$

In this formulation, $\mu_{tm}(t=1)$, $\mu_{tm}(t \geq 2)$, and $\theta_{tm}(t \geq 1)$ are the Lagrange multipliers of the constraints in Equations (2)–(4), respectively, while all $\mu_{tm}$ are grouped to $T \times M$ dimension column vector $\mu$ and $\theta_{tm}$ are grouped to $(T-1) \times M$ dimension column vector $\theta$. Note that trading prices $p_{tmn}^{1*}$ and $p_{tmn}^{u*}$ are not considered variables, rather, they are treated as endogenous variables whose value can be determined by the solution of the model.

Equation (5) has a good economic explanation. In particular, the fourth term of the sum formula indicates that when the manufacturer's optimal green degree is greater than zero, the marginal subsidy income of green degree equals the sum of the marginal costs. Similarly, when the raw material collection amount, the waste material recycling amount, the product transaction volume, the used material trading volume, and the inventory quantity have positive optimum values, the corresponding marginal revenue is equal to the marginal cost.

### 4.2. Behavior of the Retailers and Their Equilibrium Conditions

The decision of retailer n will involve deciding the quantity of product and used materials traded with manufacturers, the volume of product and used product traded with customers in all markets, the inventory in every period, and the green degree of its supply chain, which aims to maximize its profit. Its decision model can be expressed as

$$
\begin{aligned}
\max \pi_n(q_n^1, q_n^2, q_n^3, q_n^4, I_n^N, \delta_n) = & \sum_{t=1}^{T} \frac{1}{(1+\gamma)^t} \left\{ \begin{array}{l} \sum_{k=1}^{K} \left[ p_{tnk}^2 q_{tnk}^2 - C_{tnk}^{rN}(Q_t^2, \delta_n) \right] - \sum_{m=1}^{M} \left[ p_{tmn}^1 q_{tmn}^1 + C_{tmn}^{rN}(Q_t^1, \delta_n) \right] \\ -C_{tn}(Q_t^2, \delta_n) - H_{tn}(I_t^N, \delta_n) + \omega_n \delta_n \sum_{k=1}^{K} q_{tnk}^2 \end{array} \right\} \\
& + \sum_{t=2}^{T} \frac{1}{(1+\gamma)^t} \left\{ \sum_{m=1}^{M} \left[ p_{tmn}^4 q_{tmn}^4 - C_{tmn}^{uN}(Q_t^4, \delta_n) \right] - \sum_{k=1}^{K} \left[ p_{tnk}^3 q_{tnk}^3 + C_{tnk}^{uN}(Q_t^3, \delta_n) \right] - C_{tn}^u(Q_t^3, \delta_n) - \overline{\rho}(1-\chi) \sum_{k=1}^{K} q_{tnk}^3 \right\} - V_n(\delta^N)
\end{aligned}
\tag{6}
$$

s.t:

$$
\sum_{m=1}^{M} q_{tmn}^1 \geq I_{tn}^N + \sum_{k=1}^{K} q_{tnk}^2 , \ t = 1
\tag{7}
$$

$$
I_{(t-1)n}^N + \sum_{m=1}^{M} q_{tmn}^1 \geq I_{tn}^N + \sum_{k=1}^{K} q_{tnk}^2 , t \in \{2, \cdots, T\}
\tag{8}
$$

$$\chi \sum_{k=1}^{K} q_{tnk}^{3} \geq \sum_{m=1}^{M} q_{tmn}^{4}, t \in \{2, \cdots, T\} \tag{9}$$

$$\delta_m \geq \delta_n \tag{10}$$

$$\delta_n \geq \underline{\delta_n} \tag{11}$$

where $q_{tmn}^{1}, q_{tmn}^{4}, q_{tnk}^{2}, q_{tnk}^{3}, I_{tn}^{N}, \delta_n \geq 0, \forall t \in \{1, \cdots, T\}, m \in \{1, \cdots, M\}, n \in \{1, \cdots, N\}, k \in \{1, \cdots, K\}$.

The profit of retailer n is the sum of product sales revenue $\sum_{k=1}^{K} p_{tnk}^{2} q_{tnk}^{2}$, used materials' sales revenue $\sum_{m=1}^{M} p_{tmn}^{4} q_{tmn}^{4}$ and subsidies $\omega_n \delta_n \sum_{k=1}^{K} q_{tnk}^{2}$ minus various costs, which include: transaction costs $C_{tmn}^{rN}(Q_t^1, \delta_n), C_{tnk}^{rN}(Q_t^2, \delta_n), C_{tmn}^{uN}(Q_t^4, \delta_n),$ and $C_{tnk}^{uN}(Q_t^3, \delta_n)$; products purchasing costs $\sum_{m=1}^{M} p_{tmn}^{1} q_{tmn}^{1}$; waste product repurchase costs $\sum_{k=1}^{K} p_{tnk}^{3} q_{tnk}^{3}$; products exhibition and advertising expenses $C_{tn}(Q_t^2, \delta_n)$; waste product disassembling, cleaning and picking costs $C_{tn}^{u}(Q_t^3, \delta_n)$; inventory costs $H_{tn}(I_t^{N}, \delta_n)$; and disposal fees for unused waste products $\bar{\rho}(1 - \beta_u) \sum_{n=1}^{N} q_{tmn}^{4}$. The sum of discounted profits of every period less the initial GSC investment obtains the gross profit of retailer n.

Equations (7)–(9) are flow conservation constraints. Equation (7) means the sum of sales volume and inventory will not exceed the order quantity in the first period. However, from the second period, the products sold may come from the previous inventory in addition to the current order. Thus, Equation (8) specifies that the retailer's sales volume and current inventory should not exceed the sum of the order quantity and the previous inventory. The constraint of Equation (9) states that there is a maximum conversion rate from recycled products to reusable materials. The constraint of Equation (10) means retailers have green preferences and only choose manufacturers whose greenness is not less than that of the retailers themselves. The inequality of Equation (11) indicates that retailers' greenness should not be less than a certain minimum in order to meet the government's environmental regulatory requirements.

Similar to manufacturers, assuming non-cooperative competition among retailers, the Nash equilibrium of the retail market is shown as satisfying the following variational inequalities.

Determine $(Q^{1*}, Q^{2*}, Q^{3*}, Q^{4*}, I^{N*}, \delta^{N*}, \lambda^*, \xi^*, \eta^*, \nu^*) \in R^{TMN+TNK+(T-1)NK+(T-1)MN+TN+N+TN+(T-1)N+MN+N}$ satisfying

$$
\begin{aligned}
&\sum_{t=1}^{T} \sum_{m=1}^{M} \sum_{n=1}^{N} \left\{ \frac{1}{(1+\gamma)^t} \left[ \partial C_{tmn}^{rN}(Q_t^{1*}, \delta_n^*) / \partial q_{tmn}^1 + p_{tmn}^{1*} \right] - \lambda_{tn}^* \right\} (q_{tmn}^1 - q_{tmn}^{1*}) \\
&+ \sum_{t=1}^{T} \sum_{n=1}^{N} \sum_{k=1}^{K} \left\{ \frac{1}{(1+\gamma)^t} \left[ \partial C_{tnk}^{rN}(Q_t^{2*}, \delta_n^*) / \partial q_{tnk}^2 + \partial C_{tn}(Q_t^{2*}, \delta_n^*) / \partial q_{tnk}^2 - \omega_n \delta_n^* - p_{tnk}^{2*} \right] + \lambda_{tn}^* \right\} (q_{tnk}^2 - q_{tnk}^{2*}) \\
&+ \sum_{t=2}^{T} \sum_{n=1}^{N} \sum_{k=1}^{K} \left\{ \frac{1}{(1+\gamma)^t} \left[ \partial C_{tn}^{u}(Q_t^{3*}, \delta_n^*) / \partial q_{tnk}^3 + \partial C_{tnk}^{uN}(Q_t^{3*}, \delta_n^*) / \partial q_{tnk}^3 + p_{tnk}^{3*} + \bar{\rho}(1-\chi) \right] - \chi \xi_{tn}^* \right\} (q_{tnk}^3 - q_{tnk}^{3*}) \\
&+ \sum_{t=2}^{T} \sum_{m=1}^{M} \sum_{n=1}^{N} \left\{ \frac{1}{(1+\gamma)^t} \left[ \partial C_{tmn}^{uN}(Q_t^{4*}, \delta_n^*) / \partial q_{tmn}^4 - p_{tmn}^{4*} \right] + \xi_{tn}^* \right\} (q_{tmn}^4 - q_{tmn}^{4*}) \\
&+ \sum_{t=1}^{T-1} \sum_{n=1}^{N} \left[ \frac{1}{(1+\gamma)^t} \partial H_{tn}(I_t^{N*}, \delta_n^*) / \partial I_{tn}^N + \lambda_{tn}^* - \lambda_{(t+1)n}^* \right] (I_{tn}^N - I_{tn}^{N*}) + \sum_{n=1}^{N} \left[ \frac{1}{(1+\gamma)^T} \partial H_{Tn}(I_T^{N*}, \delta_n^*) / \partial I_{Tn}^N + \lambda_{Tn}^* \right] (I_{Tn}^N - I_{Tn}^{N*}) \\
&+ \sum_{t=1}^{T-1} \sum_{n=1}^{N} \left[ \frac{1}{(1+\gamma)^t} \partial H_{tn}(I_t^{N*}, \delta_n^*) / \partial I_{tn}^N + \lambda_{tn}^* - \lambda_{(t+1)n}^* \right] (I_{tn}^N - I_{tn}^{N*}) + \sum_{n=1}^{N} \left[ \frac{1}{(1+\gamma)^T} \partial H_{Tn}(I_T^{N*}, \delta_n^*) / \partial I_{Tn}^N + \lambda_{Tn}^* \right] (I_{Tn}^N - I_{Tn}^{N*}) \\
&+ \sum_{n=1}^{N} \left[ \sum_{m=1}^{M} q_{1mn}^{1*} - I_{1n}^{N*} - \sum_{k=1}^{K} q_{1nk}^{2*} \right] [\lambda_{1n} - \lambda_{1n}^*] + \sum_{t=2}^{T} \sum_{n=1}^{N} \left[ I_{(t-1)n}^{N*} + \sum_{m=1}^{M} q_{tmn}^{1*} - I_{tn}^{N*} - \sum_{k=1}^{K} q_{tnk}^{2*} \right] [\lambda_{tn} - \lambda_{tn}^*] \\
&+ \sum_{t=2}^{T} \sum_{n=1}^{N} \left[ \chi \sum_{k=1}^{K} q_{tnk}^{3*} - \sum_{m=1}^{M} q_{tmn}^{4*} \right] [\xi_{tn} - \xi_{tn}^*] + \sum_{m=1}^{M} \sum_{n=1}^{N} [\delta_m^* - \delta_n^*][\eta_{mn} - \eta_{mn}^*] + \sum_{n=1}^{N} [\delta_n^* - \underline{\delta_n}][\nu_n - \nu_n^*] \geq 0, \\
&\forall (Q^1, Q^2, Q^3, Q^4, I^N, \delta^N, \lambda, \xi, \eta, \nu) \in R^{TMN+TNK+(T-1)NK+(T-1)MN+TN+N+TN+(T-1)N+MN+N}.
\end{aligned} \tag{12}
$$

In the equation, $\lambda_{tn}(t = 1)$ and $\lambda_{tn}(t \geq 2)$ are Lagrange multipliers of the constraint conditions of Equations (7) and (8), respectively. All $\lambda_{tn}(t \geq 1)$ forms a $T \times N$ dimension column vector $\lambda$. $\xi_{tn}$, and $v_n$ are Lagrange multipliers of the constraints in Equations (9) and (11), respectively, which are grouped to $(T-1) \times N$ dimension column vectors $\xi$ and $N$ dimension column vectors $v$. $\eta_{mn}$ is the Lagrange multiplier of the constraint condition in Equation (10) and is classified into an $M \times N$ dimension column vector $\eta$. Trading prices $p_{tnk}^{2*}$ and $p_{tnk}^{3*}$ are also endogenous variables whose values can be fixed from the solution of the model.

Similar to Equation (5), Equation (12) also has good economic explanations. In particular, the sixth term of Equation (12) shows that when the optimal greenness of retailers is greater than zero, the marginal subsidy income related to greenness equals the marginal cost. Similar explanations are given for the determination of the optimal product volume, the optimal waste product volume, the optimal waste material volume, and the optimal inventory volume. That is to say, when the corresponding variable reaches the optimal value greater than zero, the marginal revenue must be equal to the marginal cost.

### 4.3. Behavior of the Consumer Markets and Recycling Markets and Their Equilibrium Conditions

Any demand and recycling market must decide: (1) how many products are purchased from retailers; (2) what prices are consumers willing to pay for the products; and (3) how many waste products are returned to retailers. Similar to the spatial price equilibrium model proposed by Nagurney [37] and the Reference [47], the equilibrium conditions of consumer markets can be characterized by the following equations:

$$\frac{1}{(1+\gamma)^t}\left[p_{tnk}^{2*} + C_{tnk}(Q_t^{2*}, \delta_n^*)\right] \begin{cases} = \frac{1}{(1+\gamma)^t}p_{tnk}^{5*}, q_{tnk}^{2*} \; 0 \\ \geq \frac{1}{(1+\gamma)^t}p_{tnk}^{5*}, q_{tnk}^{2*} = 0 \end{cases}, t \in \{1, \cdots, T\} \tag{13}$$

$$D_{tnk}(p_t^{5*}, \delta_n^*) \begin{cases} = q_{tnk}^{2*}, p_{tnk}^{5*} \; 0 \\ \leq q_{tnk}^{2*}, p_{tnk}^{5*} = 0 \end{cases}, t \in \{1, \cdots, T\} \tag{14}$$

$$\frac{1}{(1+\gamma)^t}\alpha_{tnk}(Q_t^{3*}, \delta_n^*) \begin{cases} = \frac{1}{(1+\gamma)^t}p_{tnk}^{3*}, q_{tnk}^{3*} \; 0 \\ \geq \frac{1}{(1+\gamma)^t}p_{tnk}^{3*}, q_{tnk}^{3*} = 0 \end{cases}, t \in \{2, \cdots, T\} \tag{15}$$

$$\sum_{n=1}^{N} q_{tnk}^{3*} \leq l_{tk} \sum_{n=1}^{N} q_{(t-1)nk}^{2*}, t \in \{2, \cdots, T\}. \tag{16}$$

Equation (13) shows that when consumers buy products from retailers, the sum of the retailer's product price and transaction cost cannot be greater than the product price consumers are willing to pay. Equation (14) shows that when the equilibrium price is greater than zero, the volume of product transactions equals the volume of product demand. Equation (15) means that when the optimal recovery of waste products is positive, the recovery price must be equal to the negative effect of the consumer's return of waste products. Equation (16) indicates that the recovery of waste products is limited by the maximum recovery rate.

The three equations above are typical complementarity problems, which are equivalent to three variational inequalities, respectively. Combining the consumer behavior of forward and reverse chains, the equilibrium conditions of all demand and recycling markets can be characterized by the following variational inequalities: Decision $(Q^{2*}, P^{5*}, Q^{3*}, \tau^*) \in R_+^{TNK+TNK+(T-1)NK+(T-1)K}$ satisfying

$$\begin{aligned}
&\sum_{t=1}^{T-1}\sum_{n=1}^{N}\sum_{k=1}^{K}\left\{\frac{1}{(1+\gamma)^t}\left[p_{tnk}^{2*} + C_{tnk}(Q_t^{2*}, \delta_n^*) - p_{tnk}^{5*}\right] - l_{(t+1)k}\tau_{(t+1)k}^*\right\}[q_{tnk}^2 - q_{tnk}^{2*}] \\
&+ \sum_{n=1}^{N}\sum_{k=1}^{K}\frac{1}{(1+\gamma)^T}\left[p_{Tnk}^{2*} + C_{Tnk}(Q_T^{2*}, \delta_n^*) - p_{Tnk}^{5*}\right][q_{Tnk}^2 - q_{Tnk}^{2*}] + \sum_{t=1}^{T}\sum_{n=1}^{N}\sum_{k=1}^{K}\left[q_{tnk}^{2*} - D_{tnk}(p_t^{5*}, \delta_n^*)\right][p_{tnk}^5 - p_{tnk}^{5*}] \\
&+ \sum_{t=2}^{T}\sum_{n=1}^{N}\sum_{k=1}^{K}\left\{\frac{1}{(1+\gamma)^t}\left[\alpha_{tnk}(Q_t^{3*}, \delta_n^*) - p_{tnk}^{3*}\right] + \tau_{tk}^*\right\}[q_{tnk}^3 - q_{tnk}^{3*}] + \sum_{t=2}^{T}\sum_{k=1}^{K}\left[l_{tk}\sum_{n=1}^{N}q_{(t-1)nk}^{2*} - \sum_{n=1}^{N}q_{tnk}^{3*}\right][\tau_{tk} - \tau_{tk}^*] \geq 0, \\
&\forall(Q^2, P^5, Q^3, \tau) \in R_+^{TNK+TNK+(T-1)NK+(T-1)K}.
\end{aligned} \tag{17}$$

In the equation, $\tau_{tk}$ is the Lagrange multiplier of the constraint condition in Equation (16), and all $\tau_{tk}(t=2,\cdots,T;k=1,\cdots,K)$ group to $(T-1)\times K$ dimension column vector $\tau$.

### 4.4. Multiphase Closed-Loop Supply Chain Network Equilibrium Model

When the supply chain network reaches equilibrium, the amount of willingness to allocate should be equal to the amount of willingness to receive. In other words, network equilibrium should satisfy Equations (5), (12), and (17) simultaneously, which is equivalent to the equilibrium condition of a closed-loop supply chain network obtained by adding the three equations.

**Theorem 1.** *The equilibrium conditions governing the multi-period closed-loop supply chain network model are equivalent to the solution of the variational inequality problem given by: Determine* $(q^{r*},q^{u*},Q^{1*},Q^{4*},Q^{2*},Q^{3*},I^{M*},\delta^{M*},I^{N*},\delta^{N*},p^{5*},\mu^*,\theta^*,\eta^*,\lambda^*,\xi^*,\nu^*,\tau^*)$ $\in$ $R_+^{(5T-1)M+2TMN+(3T-1)NK+(3T+1)N+(T-1)K}$ *satisfying*

$$
\begin{aligned}
&\sum_{t=1}^{T}\sum_{m=1}^{M}\left\{\frac{1}{(1+\gamma)^t}\partial f_{tm}^r(q_t^{r*},\beta_r,\delta_m^*)/\partial q_{tm}^r-\beta_r\mu_{tm}^*\right\}[q_{tm}^r-q_{tm}^{r*}]\\
&+\sum_{t=2}^{T}\sum_{m=1}^{M}\left[\frac{1}{(1+\gamma)^t}\partial f_{tm}^u(q_t^{u*},\beta_u,\delta_m^*)/\partial q_{tm}^u-\beta_u\mu_{tm}^*+\theta_{tm}^*\right][q_{tm}^u-q_{tm}^{u*}]\\
&+\sum_{t=1}^{T}\sum_{m=1}^{M}\sum_{n=1}^{N}\left\{\frac{1}{(1+\gamma)^t}\left[\partial C_{tmn}^{rM}(Q_t^{1*},\delta_m^*)/\partial q_{tmn}^1+\partial C_{tmn}^{rN}(Q_t^{1*},\delta_n^*)/\partial q_{tmn}^1-\omega_m\delta_m^*\right]+\mu_{tm}^*-\lambda_{tn}^*\right\}[q_{tmn}^1-q_{tmn}^{1*}]\\
&+\sum_{t=2}^{T}\sum_{m=1}^{M}\sum_{n=1}^{N}\left\{\frac{1}{(1+\gamma)^t}\left[\partial C_{tmn}^{uM}(Q_t^{4*},\delta_m^*)/\partial q_{tmn}^4+\partial C_{tmn}^{uN}(Q_t^{4*},\delta_n^*)/\partial q_{tmn}^4+\overline{\rho}(1-\beta_u)\right]-\theta_{tm}^*+\xi_{tn}^*\right\}[q_{tmn}^4-q_{tmn}^{4*}]\\
&+\sum_{t=1}^{T-1}\sum_{n=1}^{N}\sum_{k=1}^{K}\left\{\frac{1}{(1+\gamma)^t}\left[\partial C_{tnk}^{rN}(Q_t^{2*},\delta_n^*)/\partial q_{tnk}^2+\partial C_{tn}(Q_t^{2*},\delta_n^*)/\partial Y_{tn}+C_{tnk}(Q_t^{2*},\delta_n^*)-p_{tnk}^{5*}-\omega_n\delta_n^*\right]+\lambda_{tn}^*-l_{(t+1)k}\tau_{(t+1)k}\right\}(q_{tnk}^2-q_{tnk}^{2*})\\
&+\sum_{n=1}^{N}\sum_{k=1}^{K}\left\{\frac{1}{(1+\gamma)^T}\left[\partial C_{Tnk}^{rN}(Q_T^{2*},\delta_n^*)/\partial q_{Tnk}^2+\partial C_{Tn}(Q_T^{2*},\delta_n^*)/\partial q_{Tnk}+C_{Tnk}(Q_T^{2*},\delta_n^*)-p_{Tnk}^{5*}-K_n\delta_n^*\right]+\lambda_{Tn}^*\right\}[q_{Tnk}^2-q_{Tnk}^{2*}]\\
&+\sum_{t=2}^{T}\sum_{n=1}^{N}\sum_{k=1}^{K}\left\{\frac{1}{(1+\gamma)^t}\left[\partial C_{tn}^u(Q_t^{3*},\delta_n^*)/\partial q_{tnk}^3+\partial C_{tnk}^{uN}(Q_t^{3*},\delta_n^*)/\partial q_{tnk}^3+\alpha_{tnk}(Q_t^{3*},\delta_n^*)+\overline{\rho}(1-\chi)\right]-\chi\xi_{tn}^*+\tau_{tk}^*\right\}(q_{tnk}^3-q_{tnk}^{3*})\\
&+\sum_{t=1}^{T-1}\sum_{m=1}^{M}\left\{\frac{1}{(1+\gamma)^t}\partial H_{tm}(I_t^{M*},\delta_m^*)/\partial I_{tm}^M+\mu_{tm}^*-\mu_{(t+1)m}^*\right\}\times[I_{tm}^M-I_{tm}^{M*}]+\sum_{m=1}^{M}\left\{\frac{1}{(1+\gamma)^T}\partial H_{Tm}(I_T^{M*},\delta_m^*)/\partial I_{Tm}^M+\mu_{Tm}^*\right\}[I_{Tm}^M-I_{Tm}^{M*}]\\
&+\sum_{m=1}^{M}\left\{\begin{array}{l}\sum_{t=1}^{T}\frac{1}{(1+\gamma)^t}\left[\partial f_{tm}^r(q_t^{r*},\beta_r,\delta_m^*)/\partial\delta_m+\partial H_{tm}(I_t^{M*},\delta_m^*)/\partial\delta_m+\sum_{n=1}^{N}\partial C_{tmn}^{rM}(Q_t^{1*},\delta_m^*)/\partial\delta_m-\omega_m\sum_{n=1}^{N}q_{tmn}^{1*}\right]\\[6pt]+\sum_{t=2}^{T}\frac{1}{(1+\gamma)^t}\left[\partial f_{tm}^u(q_t^{u*},\beta_u,\delta_m^*)/\partial\delta_m+\sum_{n=1}^{N}\partial C_{tmn}^{uM}(Q_t^{4*},\delta_m^*)/\partial\delta_m\right]+\partial V_m(\delta^{M*})/\partial\delta_m\end{array}\right\}\times[\delta_m-\delta_m^*]\\
&+\sum_{t=1}^{T-1}\sum_{n=1}^{N}\left[\frac{1}{(1+\gamma)^t}\partial H_{tn}(I_t^{N*},\delta_n^*)/\partial I_{tn}^N+\lambda_{tn}^*-\lambda_{(t+1)n}^*\right](I_{tn}^N-I_{tn}^{N*})+\sum_{n=1}^{N}\left[\frac{1}{(1+\gamma)^T}\partial H_{Tn}(I_T^{N*},\delta_n^*)/\partial I_{Tn}^N+\lambda_{Tn}^*\right](I_{Tn}^N-I_{Tn}^{N*})\\
&+\sum_{n=1}^{N}\left\{\begin{array}{l}\sum_{t=1}^{T}\frac{1}{(1+\gamma)^t}\left[\partial C_{tn}(Q_t^{2*},\delta_n^*)/\partial\delta_n+\partial H_{tn}(I_t^{N*},\delta_n^*)/\partial\delta_n+\sum_{m=1}^{M}\partial C_{tmn}^{rN}(Q_t^{1*},\delta_n^*)/\partial\delta_n+\sum_{k=1}^{K}\partial C_{tnk}^{rN}(Q_t^{2*},\delta_n^*)/\partial\delta_n-\omega_n\sum_{k=1}^{K}q_{tnk}^{2*}\right]\\[6pt]+\sum_{t=2}^{T}\frac{1}{(1+\gamma)^t}\left[\sum_{m=1}^{M}\partial C_{tmn}^{uN}(Q_t^{4*},\delta_n^*)/\partial\delta_n+\sum_{k=1}^{K}\partial C_{tnk}^{uN}(Q_t^{3*},\delta_n^*)/\partial\delta_n+\partial C_{tn}^u(Q_t^{3*},\delta_n^*)/\partial\delta_n\right]-\nu_n+\sum_{m=1}^{M}\eta_{mn}^*+\partial V_n(\delta^{N*})/\partial\delta_n\end{array}\right\}\times[\delta_n-\delta_n^*]\\
&+\sum_{t=1}^{T}\sum_{n=1}^{N}\sum_{k=1}^{K}[q_{tnk}^{2*}-D_{tnk}(p_t^{5*},\delta_n^*)][p_{tnk}^5-p_{tnk}^{5*}]+\sum_{m=1}^{M}[\beta_r q_{1m}^{r*}-I_{1m}^{M*}-\sum_{n=1}^{N}q_{1mn}^{1*}][\mu_{1m}-\mu_{1m}^*]\\
&+\sum_{t=2}^{T}\sum_{m=1}^{M}[I_{(t-1)m}^{M*}+\beta_r q_{tm}^{r*}+\beta_u q_{tm}^{u*}-I_{tm}^{M*}-\sum_{n=1}^{N}q_{tmn}^{1*}][\mu_{tm}-\mu_{tm}^*]+\sum_{t=2}^{T}\sum_{m=1}^{M}[\sum_{n=1}^{N}q_{tmn}^4-q_{tm}^u][\theta_{tm}-\theta_{tm}^*]\\
&+\sum_{m=1}^{M}\sum_{n=1}^{N}[\delta_m-\delta_n][\eta_{mn}-\eta_{mn}^*]+\sum_{n=1}^{N}[\sum_{m=1}^{M}q_{1mn}^{1*}-I_{1n}^{N*}-\sum_{k=1}^{K}q_{1nk}^{2*}][\lambda_{1n}-\lambda_{1n}^*]+\sum_{t=2}^{T}\sum_{n=1}^{N}[I_{(t-1)n}^{N*}+\sum_{m=1}^{M}q_{tmn}^{1*}-I_{tn}^{N*}-\sum_{k=1}^{K}q_{tnk}^{2*}][\lambda_{tn}-\lambda_{tn}^*]\\
&+\sum_{t=2}^{T}\sum_{n=1}^{N}[\chi\sum_{k=1}^{K}q_{tnk}^{3*}-\sum_{m=1}^{M}q_{tmn}^{4*}][\xi_{tn}-\xi_{tn}^*]+\sum_{n=1}^{N}[\delta_n^*-\underline{\delta_n}][\nu_n-\nu_n^*]++\sum_{t=2}^{T}\sum_{k=1}^{K}[l_{tk}\sum_{n=1}^{N}q_{(t-1)nk}^{2*}-\sum_{n=1}^{N}q_{tnk}^{3*}][\tau_{tk}-\tau_{tk}^*]\geq 0,\\
&\forall(q^r,q^u,Q^1,Q^4,Q^2,Q^3,I^M,\delta^M,I^N,\delta^N,p^5,\mu,\theta,\eta,\lambda,\xi,\nu,\tau)\in R_+^{(5T-1)M+2TMN+(3T-1)NK+(3T+1)N+(T-1)K}
\end{aligned}
\tag{18}
$$

**Theorem 2.** *Similarly to Reference [42], the transaction price between manufacturer and retailer can be obtained from the variational inequality in Equation (6) using the complementary theorem*

$$
p_{tmn}^{1*}=\partial C_{tmn}^{rM}(Q_t^{1*},\delta_m)/\partial q_{tmn}^1-\omega_m\delta_m+(1+\gamma)^t\mu_{tm}^*.
\tag{19}
$$

In the same way, the transaction price of waste products between manufacturers and retailers can be obtained from the variational inequality in Equation (12)

$$p_{tmn}^{4*} = \partial C_{tmn}^{uN}(Q_t^{4*}, \delta_n) / \partial q_{tmn}^4 + (1+\gamma)^t \xi_{tn}^*. \tag{20}$$

From the variational inequality of Equation (12), the retailer's selling price can be obtained by

$$p_{tnk}^{2*} = \partial C_{tnk}^{rN}(Q_t^{2*}, \delta_n) / \partial q_{tnk}^2 + \partial C_{tn}(Q_t^{2*}, \delta_n^*) / \partial q_{tnk}^2 - \omega_n \delta + (1+\gamma)^t \lambda_{tn}^*. \tag{21}$$

From the variational inequality of Equation (17), the transaction price of waste products between retailers and consumers in the demand market can be obtained by

$$p_{tnk}^{3*} = \alpha_{tnk}(Q_t^{3*}, \delta_n) + (1+\gamma)^t \tau_{tk}^*. \tag{22}$$

When the cost is a continuous differentiable convex function of the corresponding quantity, the demand function is a monotonic decreasing function of the demand price, and the negative return utility of the waste products of consumers is a monotonic increasing function of the return amount. Thus, the existence of the solution of the variational inequality model in Equation (18) can be proved similar to that in the Reference [5,42], which will not be discussed here.

## 5. Numerical Analysis

In this section, we: provide some numerical examples to illustrate the model; investigate the impact of increased government subsidy rates, competition intensification caused by the increase of network members and asymmetric costs on equilibrium outcomes; and also discuss the results. The projection contraction algorithm has the advantage of an adaptive step size, so it was used to solve all the following examples [48]. The convergence criterion used was that the absolute value of results between two successive iterations differed by no more than $10^{-8}$.

**Example 1.** *Consider one supply chain consisting of two manufacturers, two retailers, two demand/recycling markets, competing over five periods ($M = N = K = 2$, $T = 5$). The parameters are set as: $\gamma = 0.03$, $\beta_r = 0.8$, $\beta_u = 0.6$, $\chi = 0.8$, $l_{tk} = 0.9$, $\underline{\delta_n} = 0.1$, $\omega_m = \omega_n = 0 : 1 : 10$. $m'$, $n'$, $k'$ are the serial numbers different from $m$, $n$, and $k$ respectively.*

The manufacturing cost functions are

$$f_{tm}^r(q_t^r, \beta_r, \delta_m) = 9\left(\widetilde{q}_{tm}^r\right)^2 + a_t \widetilde{q}_{tm}^r + b_t \delta_m \widetilde{q}_{tm}^r, \ m = 1, 2, t = 1, \cdots 5.$$
$$\widetilde{q}_{tm}^r = \beta_r\left(q_{tm}^r + 0.1\sum_{m'} q_{tm'}^r\right), \ a_t = [2, 2.4, 2.8, 3.2, 3.6], \ b_t = [0.8, 0.4, 0.1, -0.2, -0.4]. \tag{23}$$

The remanufacturing cost functions are

$$f_{tm}^u(q_t^u, \beta_u, \delta_m) = 7\left(\widetilde{q}_{tm}^u\right)^2 + c_t \widetilde{q}_{tm}^u + d_t \widetilde{q}_{tm}^u, \ m = 1, 2, t = 2, \cdots 5.$$
$$\widetilde{q}_{tm}^u = \beta_u\left(q_{tm}^u + 0.1\sum_{m'} q_{tm'}^u\right), c_t = [2.5, 3, 3.3, 3.5], \ d_t = [0.6, 0.3, -0.1, -0.3]. \tag{24}$$

The product transaction costs assumed by manufacturers and retailers, respectively, are

$$C_{tmn}^{rM}(Q_t^1, \delta_m) = 4.5\left(\widetilde{q}_{tmn}^1\right)^2 + 1.8\widetilde{q}_{tmn}^1 + e_t \delta_m \widetilde{q}_{tmn}^1, C_{tmn}^{rN}(Q_t^1, \delta_n) = 5\left(\widetilde{q}_{tmn}^1\right)^2 + 3\widetilde{q}_{tmn}^1 + e_t \delta_m \widetilde{q}_{tmn}^1,$$
$$m = 1, 2, n = 1, 2, t = 1, \cdots 5. \tag{25}$$
$$\widetilde{q}_{tmn}^1 = q_{tmn}^1 + 0.2q_{tmn'}^1 + 0.1\sum_{m'}\sum_n q_{tm'n'}^1, \ e_t = [0.8, 0.4, 0.1, -0.4, -0.6].$$

The reusable material transaction costs assumed by manufacturers and retailers, respectively, are

$$C_{tmn}^{uM}(Q_t^4, \delta_m) = 4.8\left(\widetilde{q}_{tmn}^4\right)^2 + 2.3\widetilde{q}_{tmn}^4 + j_t\delta_m\widetilde{q}_{tmn}^4, \ C_{tmn}^{uN}(Q_t^4, \delta_n) = 4.7\left(\widetilde{q}_{tmn}^4\right)^2 + 2.3\widetilde{q}_{tmn}^4 + j_t\delta_m\widetilde{q}_{tmn}^4,$$
$$m = 1, 2, n = 1, 2, t = 2, \cdots 5.$$
$$j_t = [0.6, 0.3, 0.1, -0.3], \ \widetilde{q}_{tmn}^4 = q_{tmn}^4 + 0.2q_{tmn'}^4 + 0.1\sum_{m'}\sum_n q_{tm'n}^4.$$

(26)

The product transaction costs assumed by retailers and consumers are, respectively

$$C_{tnk}^{rN}(Q_t^2, \delta_n) = 4.4\left(\widetilde{q}_{tnk}^2\right)^2 + 2.2\widetilde{q}_{tnk}^2 + b_t\delta_n\widetilde{q}_{tnk}^2, \ C_{tnk}(Q_t^2) = 2, \ n = 1, 2, k = 1, 2, t = 1, \cdots 5.$$
$$b_t = [0.8, 0.4, 0.1, -0.2, -0.4], \ \widetilde{q}_{tnk}^2 = q_{tnk}^2 + 0.2q_{tnk'}^2 + 0.1\sum_{n'}\sum_k q_{tn'k}^2.$$

(27)

The waste product transaction costs assumed by retailers are

$$C_{tnk}^{uN}(Q_t^{3*}, \delta_n) = 4.6\left(\widetilde{q}_{tnk}^3\right)^2 + 2.3\widetilde{q}_{tnk}^3 + j_t\delta_n\widetilde{q}_{tnk}^3, \ m = 1, 2, n = 1, 2, t = 2, \cdots 5.$$
$$j_t = [0.6, 0.3, 0.1, -0.3], \ \widetilde{q}_{tnk}^3 = q_{tnk}^3 + 0.2q_{tnk'}^3 + 0.1\sum_{n'}\sum_k q_{tn'k}^3.$$

(28)

The cost of disassembling, cleaning and picking of recycled products assumed by retailers are

$$C_{tn}^u(Q_t^3, \delta_n) = 3.15\left(\sum_{k=1}^K \widetilde{q}_{tnk}^3\right)^2 + 2.3\sum_{k=1}^K \widetilde{q}_{tnk}^3 + j_t\delta_n\sum_{k=1}^K \widetilde{q}_{tnk}^3, \ n = 1, 2, t = 2, \cdots 5.$$
$$j_t = [0.6, 0.3, 0.1, -0.3], \ \widetilde{q}_{tnk}^3 = q_{tnk}^3 + 0.2q_{tnk'}^3 + 0.1\sum_{n'}\sum_k q_{tn'k}^3$$

(29)

Retailers' product exhibition and advertising expenses are

$$C_{tn}(Q_t^2, \delta_n) = u_t\left(\hat{q}_{tn}^2\right)^2 + v_t\hat{q}_{tn}^2 + w_t\delta_n\hat{q}_{tn}^2, \ m = 1, 2, n = 1, 2, t = 1, \cdots 5.$$
$$C_{tn}(Q_t^2, \delta_n) = u_t\left(\hat{q}_{tn}^2\right)^2 + v_t\hat{q}_{tn}^2 + w_t\delta_n\hat{q}_{tn}^2, \ u_t = [2.9, 2.8, 2.7, 2.6, 2.5],$$
$$v_t = [2.8, 2.6, 2.4, 2.2, 2.1], \ w_t = [0.3, 0.1, -0.1, -0.3, -0.35].$$

(30)

The inventory costs assumed by manufacturers and retailers are, respectively

$$H_{tm}(I_t^M, \delta_m) = 2.6\left(\widetilde{I}_{tm}^M\right)^2 + 1.3\widetilde{I}_{tm}^M + h_t\delta_m\widetilde{I}_{tm'}^M, \ H_{tn}(I_t^N, \delta_n) = 2.5\left(\widetilde{I}_{tn}^N\right)^2 + 1.2\widetilde{I}_{tn}^N + h_t\delta_n\widetilde{I}_{tn'}^N,$$
$$m = 1, 2, n = 1, 2, t = 1, \cdots 5. \widetilde{I}_{tm}^M = I_{tm}^M + 0.2\sum_{m'} I_{tm''}^M, \ \widetilde{I}_{tn}^N = I_{tn}^N + 0.2\sum_{n'} I_{tn''}^N,$$
$$h_t = [0.3, 0.1, -0.1, -0.3, -0.5].$$

(31)

The consumers' negative utility for returning used products is

$$\alpha_{tnk}(Q_t^3, \delta_n) = 15q_{tnk}^3 - s_t\delta_n\hat{q}_{tnk}^3, \ n = 1, 2, k = 1, 2, t = 2, \cdots 5.$$
$$s_t = [1, 0.9, 0.8, 0.7], \ \hat{q}_{tnk}^3 = q_{tnk}^3 + 0.1q_{tn'k}^3.$$

(32)

The GSC technology investment of manufacturers and retailers is, respectively

$$V_m(\delta^M) = 30 + 40\left(\widetilde{\delta}_m\right)^2, \ V_n(\delta^N) = 20 + 30\left(\widetilde{\delta}_n\right)^2, \ m = 1, 2, n = 1, 2$$
$$\widetilde{\delta}_m = \delta_m + 0.1\sum_{m'}\delta_{m'}, \ \widetilde{\delta}_n = \delta_n + 0.1\sum_{n'}\delta_{n'}.$$

(33)

The demand functions are

$$D_{tnk}(p_t^5, \delta_n) = g_t - f_tp_{tnk}^5 + r_t\delta_n + 0.05p_{tn'k'}^5, \ n = 1, 2, k = 1, 2, t = 1, \cdots 5.$$
$$g_t = [210, 220, 230, 240, 250], \ f_t = [0.9, 0.8, 0.7, 0.6, 0.5], \ r_t = [-2, 0, 2, 4, 5].$$

(34)

The results of Table 2, Figures 4 and 5 were obtained using the MATLAB program.

Figures 4 and 5 show that when the government does not grant subsidies, enterprises have no incentive to invest in GSC technology, and the greenness of enterprises remains at the lowest level of 0.1. However, with the continuous increase of the government subsidy rate (GSR), single retailer's technology investment in GSC (SRTI) and single manufacturer's technology investment in GSC (SMTI) keep increasing, the green rate of manufacturers (GRM) and the green rate of retailers (GRR) keeps increasing, the total supply of products (TSP) in the market increases, and the total amount of government subsidies (TAGS) keeps rising. Figure 5a,b show that as the government subsidy rate increases, the company's income and costs rise, but the increase in income exceeds the cost increase, which makes the company's profits continue to rise. In reality, the government is bound by funds. The subsidy of the enterprise must be limited to a certain range. For example, setting the government subsidy rate to 4 to 6 is a relatively appropriate level, and the ratio of subsidies to corporate profits is also reasonable. Figure 5c shows that profit rates are declining as profits rise faster than costs. Due to the enhancement of public environmental awareness, the demand for greener products is increasing, and the relative cost of using recycled materials is lower than using raw materials to produce products. Enterprises will inevitably improve the recovery rate of waste products. Figure 5d confirms this conclusion. From the above analysis, we draw the following conclusion:

**Conclusion 1.** *Government subsidies can effectively encourage enterprises to invest in GSC technology and improve their profits and social welfare.*

In order to reflect the influence of network structure change on the equilibrium results, a manufacturer is added on the basis of Example 1.

**Example 2.** *Consider a supply chain network composed of three producers, two retailers, and two demand/recycling markets. The parameters and cost functions of the new manufacturer are the same as those of Example 1. In particular, the setting of the transaction cost function between new manufacturers and retailers is similar to that of two other original manufacturers. The calculation results are shown in Table 2.*

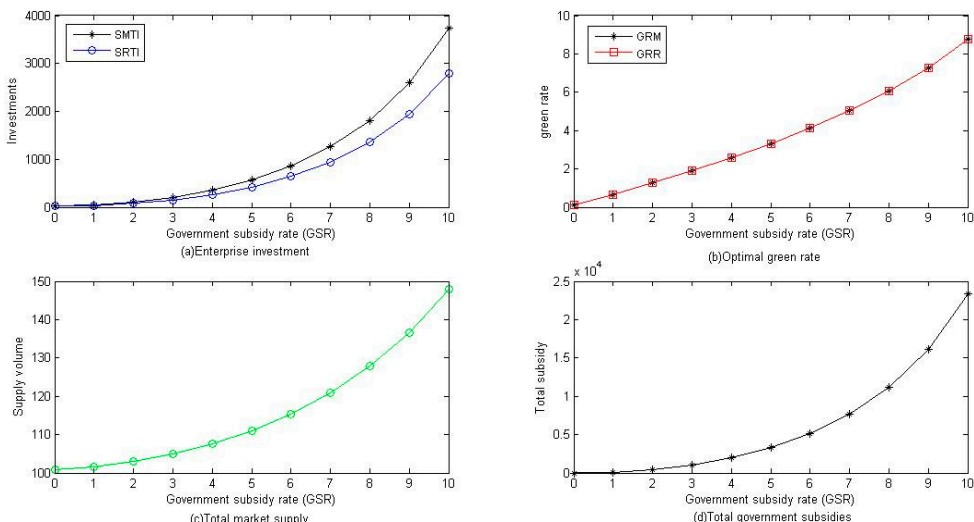

**Figure 4.** The role of government subsidies in the promotion of green technology investment. Notes: In Figure 4 and the following images and tables, the government subsidy rate is denoted as GSR, single retailer's technology investment in GSC is denoted as SRTI, single manufacturer's technology investment in GSC is denoted as SMTI, the green rate of manufacturers is denoted as GRM, the green rate of retailers is denoted as GRR, the total supply of products is denoted as TSP, and the total amount of government subsidies is denoted as TAGS.

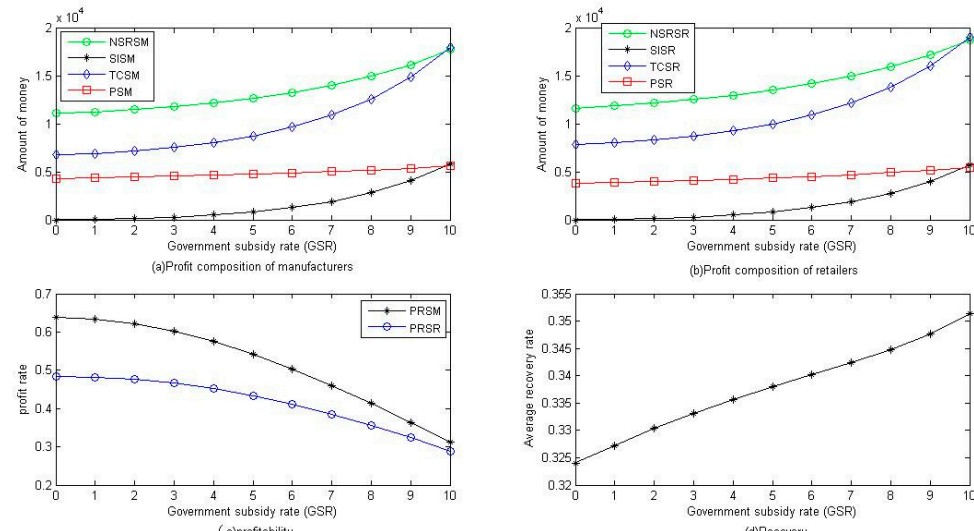

**Figure 5.** Influence of government subsidy on enterprise profit and recovery rate. Notes: In Figure 5 and the following images and tables, net sales revenue of a single manufacturer is denoted as NSRSM, subsidy income of a single manufacturer is denoted as SISM, total cost of a single manufacturer is denoted as TCSM, profit of a single manufacturer is denoted as PSM, net sales revenue of a single retailer is denoted as NSRSR, subsidy income of a single retailer is denoted as SISR, total cost of a single retailer is denoted as TCSR, profit of a single retailer is denoted as PSR, profit rate of a single manufacturer is denoted as PRSM, profit rate of a single retailer is denoted as PRSR, and average recycling rate of a single retailer is denoted as ARRSR. Net sales revenue of a single manufacturer (NSRSM) = sales revenue of products-purchase cost of reusable materials, net sales revenue of a single retailer (NSRSR) = sales revenue of products-purchase cost of products + sales revenue of reusable materials-purchase cost of waste products. Profit rate (PRSM or PRSR) = enterprise profit/total cost.

Table 2 shows that after adding a new manufacturer, the supply of products in the market increases substantially, the retail price of products decreases slightly, the optimal green degree of enterprises decreases, the manufacturer's profit decreases, and the retailer's profit rises sharply. With the addition of a new manufacturer, the competition in production and manufacture of products intensifies, and the number of retailers remains unchanged, which leads to retailers being in a more advantageous position relative to manufacturers and gaining more profits than the original market structure.

Based on the above analysis, we summarize the following conclusion:

**Conclusion 2.** *The intensification of competition in the supply market will weaken the incentive effect of government subsidies, reduce product prices, and improve retailer profits and consumer welfare.*

In order to reflect the impact of the unequal strength of enterprises on the equilibrium results, two manufacturers have been set different cost functions.

**Example 3.** *In order to reflect the influence of differences in firm strengths, based on Example 1, we reset the manufacturing cost, remanufacturing cost, and green technology investment cost of manufacturer 1 to make it smaller than that of manufacturer 2, which reflects that manufacturer 1 has promoted technology for the above processes. However, for simplicity, the manufacturer 1 transaction cost setting remains the same as in Example 1. In addition. the cost functions and parameters of manufacturer 2 and retailers are the same as those of Example 1.*

The cost of production, remanufacturing cost, and technology investment function of GSC of manufacturer 1 are set as follows (m = 1):

$$f_{tm}^r(q_{tm}^r, \beta_r, \delta_m) = 8\left(\beta_r q_{tm}^r\right)^2 + \widetilde{a}_t \beta_r q_{tm}^r + b_t \delta_m q_{tm}^r, \ \widetilde{a}_t = [1.8, 2.2, 2.6, 3, 3.4], \ b_t = [0.8, 0.4, 0.1, -0.2, -0.4] \tag{35}$$

$$f_{tm}^u(q_{tm}^u, \beta_u, \delta_m) = 6\left(\beta_u q_{tm}^u\right)^2 + \widetilde{c}_t \beta_u q_{tm}^u + d_t \delta_m q_{tm}^u, \ \widetilde{c}_t = [2, 2.5, 3, 3.3], \ d_t = [0.6, 0.3, -0.1, -0.3]; \quad (36)$$

$$V_m(\delta_m) = 25 + 35(\delta_m)^2. \quad (37)$$

**Table 2.** Influence of network structure change on equilibrium results.

| $K_m=K_n=a$ | | $a$=0 | $a$=1 | $a$=2 | $a$=3 | $a$=4 | $a$=5 | $a$=6 | $a$=7 | $a$=8 | $a$=9 | $a$=10 |
|---|---|---|---|---|---|---|---|---|---|---|---|---|
| TSP | I | 100.8139 | 101.5674 | 102.9350 | 104.9210 | 107.5950 | 111.0576 | 115.4512 | 120.9769 | 127.9212 | 136.7019 | 147.9473 |
| | II | 121.2910 | 122.0623 | 123.4981 | 125.5791 | 128.3686 | 131.9568 | 136.4693 | 142.0793 | 149.0269 | 157.6497 | 168.4327 |
| $P_{ER}$ | I | 555.4802 | 560.4887 | 566.0229 | 571.7340 | 577.7315 | 584.1453 | 591.1362 | 598.9119 | 607.7518 | 618.0479 | 630.3744 |
| | II | 553.0695 | 557.2947 | 562.1229 | 567.0695 | 572.2159 | 577.6558 | 583.5020 | 589.8956 | 597.0194 | 605.1191 | 614.5372 |
| GR | I | 0.1000 | 0.6533 | 1.2703 | 1.9129 | 2.5938 | 3.3281 | 4.1350 | 5.0390 | 6.0736 | 7.2858 | 8.7446 |
| | II | 0.1000 | 0.5694 | 1.1116 | 1.6734 | 2.2644 | 2.8957 | 3.5811 | 4.3377 | 5.1879 | 6.1621 | 7.3029 |
| PSM | I | 4345.54 | 4394.91 | 4461.25 | 4542.24 | 4639.20 | 4753.59 | 4886.88 | 5040.35 | 5214.59 | 5408.36 | 5615.92 |
| | II | 2614.00 | 2631.77 | 2644.86 | 2649.80 | 2644.55 | 2625.68 | 2587.78 | 2522.47 | 2416.61 | 2249.31 | 1986.31 |
| PSR | I | 3808.98 | 3886.60 | 3981.87 | 4090.43 | 4214.56 | 4356.98 | 4521.09 | 4711.15 | 4932.62 | 5192.61 | 5500.60 |
| | II | 4229.74 | 4302.77 | 4401.27 | 4519.19 | 4659.46 | 4825.96 | 5023.89 | 5260.22 | 5544.60 | 5890.61 | 6318.05 |

Notes: I denotes the network structure of Example 1 with two manufacturers, and II denotes the network structure of this example after adding one more manufacturer. Symbols TSP, GR, PSM, and PSR represent total supply of products, green rate, profit of a single manufacturer and profit of a single retailer, respectively. The retail price of product at the end of period is denoted as $P_{ER}$ in Tables 2 and 3. Because we find that the manufacturer's greenness is equal to the retailer's greenness, for the sake of simplicity, we use a symbol GR to indicate both the manufacturer's greenness and the retailer's greenness.

We define the case of Example 1 as type 1, the case of Example 2 as type 2, and the case of this example as type 3. The calculation results are shown in Table 3.

**Table 3.** Equilibrium results of asymmetric and symmetric structures.

| $K_m=K_n=a$ | | $a$=0 | $a$=1 | $a$=2 | $a$=3 | $a$=4 | $a$=5 | $a$=6 | $a$=7 | $a$=8 | $a$=9 | $a$=10 |
|---|---|---|---|---|---|---|---|---|---|---|---|---|
| $Q_j$ in type i | i=III, j = 1 | 53.5636 | 53.9879 | 54.7570 | 55.8770 | 57.3895 | 59.3557 | 61.8624 | 65.0338 | 69.0494 | 74.1756 | 80.8218 |
| | i=III, j = 2 | 49.4914 | 49.8854 | 50.5986 | 51.6365 | 53.0379 | 54.8592 | 57.1812 | 60.1190 | 63.8391 | 68.5885 | 74.7472 |
| | i=I, j = 1/2 | 50.4070 | 50.78 | 51.4675 | 52.4605 | 53.7975 | 55.5288 | 57.7256 | 60.4884 | 63.9606 | 68.3510 | 73.9736 |
| $w_{Ej}$ in type i | i=III, j = 1 | 261.33 | 263.35 | 265.68 | 268.19 | 270.96 | 274.04 | 277.52 | 281.54 | 286.25 | 291.90 | 298.83 |
| | i=III, j = 2 | 264.52 | 266.58 | 268.96 | 271.55 | 274.40 | 277.60 | 281.24 | 285.44 | 290.39 | 296.33 | 303.64 |
| | i=I, j = 1/2 | 267.22 | 269.17 | 271.45 | 273.93 | 276.67 | 279.74 | 283.22 | 287.23 | 291.92 | 297.51 | 304.31 |
| Enterprises' GR in type III | | 0.1000 | 0.6917 | 1.3476 | 2.0324 | 2.7607 | 3.5498 | 4.4221 | 5.4067 | 6.5440 | 7.8917 | 9.5367 |
| PM1 in type III | | 4579.18 | 4637.02 | 4718.49 | 4822.00 | 4950.27 | 5106.77 | 5295.98 | 5523.60 | 5796.80 | 6124.39 | 6516.44 |
| PM2 in type III | | 4144.07 | 4192.44 | 4253.67 | 4325.20 | 4407.56 | 4500.83 | 4604.31 | 4715.81 | 4830.18 | 4936.33 | 5010.63 |
| TSP in type III | | 103.05 | 103.87 | 105.36 | 107.51 | 110.43 | 114.21 | 119.04 | 125.15 | 132.89 | 142.76 | 155.57 |
| $P_{ER}$ in type III | | 555.24 | 560.60 | 566.48 | 572.56 | 578.97 | 585.85 | 593.40 | 601.86 | 611.56 | 623.00 | 636.88 |
| PSR in type III | | 3939.46 | 4023.45 | 4125.80 | 4242.46 | 4375.94 | 4529.24 | 4706.09 | 4911.09 | 5150.11 | 5430.62 | 5762.32 |

Notes: In this table, the trading volume of products between manufacturer j and all retailers is denoted as $Q_j$, and the transaction price of product between manufacturer j and retailers at the end of period is denoted as $w_{Ej}$. The profit of manufacturer 1 is denoted as PM1, and the profit of manufacturer 2 is denoted as PM2.

Although manufacturer 1 has lower manufacturing costs, remanufacturing costs, and technology investment costs than manufacturer 2, we conclude that the two manufacturers and two retailers have the same greenness and same retail price. Table 3 shows that manufacturer 1 makes full use of the advantages of lower production cost and remanufacturing cost to produce and sell more products at a lower wholesale price than manufacturer 2, and earn more profits. Because the costs of manufacturer 1 are lower than that of manufacturers of type I (Example 1), it can be seen that the technology of manufacturer 1 in type I has been improved, which also improves the average technology level of the manufacturer group. Thus, compared with Example 1, the supply increases,

but the retail price rises slightly, eventually making the type III retailers' profit higher than the type I retailers' profit. Compared to manufacturer 2 who has the same green technology investment function as in type I, manufacturer 1 in this type can achieve the same greenness with less investment, which also affects the GSC technology investment level of the overall supply chain. Ultimately, under the same conditions of government subsidies, the optimal greenness of the supply chain is higher than that of type I.

Through the analysis of this example, we draw the following conclusion:

**Conclusion 3.** *Manufacturers with a cost advantage will increase the investment in GSC technology, gain more profits, and improve the overall technology level of the supply chain.*

## 6. Discussion

Subsidies are relatively common in emerging market economies, such as China, India, Brazil, and Vietnam. In order to promote economic development, and to attract and accept industrial transfer from developed countries, these countries tolerated heavy polluting industries in the early stage. However, with the continuous development of the economies of these countries, the public's awareness of environmental protection has gradually increased. The government needs to balance economic interests and environmental impacts, eliminate or transfer heavy polluting industries, and take a series of measures to help and encourage some industries with less serious pollution and good economic benefits to improve their green level. Common subsidies include appropriation, tax allowances and exemptions, and priority and low interest loans. In addition, in international trade, export subsidies undoubtedly enhance the competitive advantage of the products with which they are associated, but which are vulnerable to countervailing measures by importing countries on the grounds of unfair competition. Green subsidies are classified as non-actionable subsidies by the Agreement on Subsidies and Countervailing Measures (SCM) implemented and allowed by the World Trade Organization (WTO), and the member who suffered from the subsidies cannot appeal to the Dispute Settlement Body (DSB) or implement corresponding countervailing measures. However, the complexity of international trade is beyond the reach of this agreement, and trade disputes caused by subsidies occur from time to time.

For developed countries, due to the mature market economic system, sound legal system of environmental protection, and high threshold of market access, some industries with heavy pollution are excluded from the country. These countries mainly subsidize the production and use of renewable energy. Many countries offer very attractive subsidies to users who purchase new energy vehicles in order to promote the popularity of these vehicles. For example, the policy introduced in spring 2016 in Germany provides a subsidy of EUR 4000 for users who purchase new electric vehicles and EUR 3000 for users who buy plug-in hybrid vehicles. The British subsidy policy is simpler and clearer. Regardless of the energy-driven mode of the vehicle, as long as the vehicle's electric endurance can exceed 70 miles, it can receive a 35% subsidy of the price, to a ceiling of EUR 4500. If the vehicle's pure electric endurance is less than 70 miles, but above 10 miles, the subsidy ceiling is reduced to EUR 2500. While other countries are offering incentives and exemptions based on energy-driven vehicles, France's incentives are based on carbon dioxide emissions from vehicles. Whether pure electric or hybrid vehicles, vehicles with carbon emissions below 20 g/km will receive a subsidy of EUR 6000. If the emission is higher than 21 g/km and less than 60 g/km, EUR 1000 will be obtained. The 'feed-in tariff' policy, which prevails in the EU, has greatly stimulated people's enthusiasm to participate in photovoltaic power generation and wind power generation. Up to now, nearly one-third of energy consumption in the EU has been provided by renewable energy sources. Accordingly, the EU has ambitiously set a target of 100% renewable energy supply for energy consumption by 2050. It is worth mentioning that the feed-in tariff scheme plans to gradually reduce until subsidies for renewable energy supply are eliminated, in order to encourage enterprises to carry out technological innovation to reduce

production and operation costs. Following this theme, our future research will consider the dynamic subsidy situation.

Governments' green subsidies play an important role in promoting green investment and green development, but also increase a government's financial burden. Some alternative schemes can also be tried, such as the new-energy vehicle credit program and corporate average fuel consumption regulation (dual-credit policy) implemented by the Chinese government to promote the development of the new energy automobile industry [49], or other indirect interventions, such as environmental taxes or the cap-and-trade policy, or the combination of these interventions.

Generally speaking, green subsidies are conducive to improving the ecological environment, and achieving sustainable development of resources and the ecological environment, which is worth advocating for both developed and developing countries. For developing countries, green subsidies also play a role in promoting the upgrade of the industrial structure, balancing economic and environmental benefits. Essentially, green subsidies are a measure to resolve the external diseconomy of pollution emissions and to promote the internalization of the social cost of enterprise pollution emissions.

## 7. Conclusions

A government may adopt a subsidy policy to promote enterprises to invest in green technology. For the first time, this paper used the method of multi-period closed-loop supply chain network equilibrium to study the problem of government subsidies for enterprises' technological investment in GSC, and analyses the effects of subsidies, enterprise cost asymmetry, and market structure changes on the equilibrium solution. In summary, we drew the following conclusions and recommendations:

The long-term decision-making of enterprises was investigated. Business operation is not a one-shot deal, and its long-term behavior is inevitably different from its short-term behavior. The technology investment of GSC may not be profitable in the short term, but can save costs in the long term. In this paper, we set different costs in different periods. In particular, the cost monotonously decreases with the green rate since some certain phase. Under this background, the long-term decision-making behavior of enterprises' GSC investment is studied.

The competition problem of the multi-tier network was considered. Many manufacturers, retailers, and markets constitute a network with a multi-tier structure, and the strength of members at the same level is often asymmetric. This paper concludes that the improvement of the operational efficiency of a single enterprise is conducive to improving the overall green level of the network.

The government subsidy under competition was studied. The research shows that government subsidies can effectively promote enterprises to upgrade the level of technology investment in GSC, and the intensification of competition at the product supply level will weaken the effect of government subsidies, which was counter to our intuition. Generally speaking, when competition intensifies, enterprises will expand their investment in GSC technology to earn more government subsidies, but this paper draws the opposite conclusion.

This paper assumed that new products and remanufactured products are homogeneous and have the same price, but—in reality—new products and remanufactured products often have different prices and levels of demand. Moreover, for simplicity, this paper assumes that the technology investment of GSC is a one-time investment while—in reality—enterprises often make dynamic and continuous investments. In view of this, in the future we can further study the heterogeneity of new products and remanufactured products, the supply chain network equilibrium model under multi-period investment, and dynamic subsidies.

**Author Contributions:** Conceptualization, H.W. and B.X.; Methodology, H.W. and B.X.; Supervision, B.X.; Writing—original draft, H.W.; Writing—review and editing, H.W., B.X., and D.Z.

**Funding:** This research was supported by National Natural Science Foundation of China (grant no. 71561018), Science and Technology Project of the Education Department of Jiangxi Province (grant no. GJJ181386), Graduate Innovation Program of Jiangxi Province (grant no. YC2018-B009).

**Conflicts of Interest:** The authors declare no conflict of interest.

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
