# Peer review of "Closed-Loop Supply Chain Network Equilibrium Model with Subsidy on Green Supply Chain Technology Investment"

_sustainability, doi:10.3390/su11164403_

Round 1
Reviewer 1 Report
This is an interesting work explaining the role of green supply chain (GSC) on technology investment. The authors have elaborated the topic very well and the results are easy to follow. At this point, the only recommendation is to add a section to explain how this model is flexible for utilization in different countries (developed and developing). A part of that is covered in this paper, but that would be great to indicate the challenges that use of such model may bring up in specific geographical locations depending on their financial culture. Other than that, the paper looks good.
Author Response
Dear Professor,
On behalf of my co-authors, we thank you very much for giving us an opportunity to revise our manuscript, we appreciate you very much for your positive and constructive comments and suggestions on our manuscript entitled “Closed-loop Supply Chain Network Equilibrium Model with Subsidy on GSC Technology Investment” (sustainability-554868).
Your comments on our article are as follows:
“This is an interesting work explaining the role of green supply chain (GSC) on technology investment. The authors have elaborated the topic very well and the results are easy to follow. At this point, the only recommendation is to add a section to explain how this model is flexible for utilization in different countries (developed and developing). A part of that is covered in this paper, but that would be great to indicate the challenges that use of such model may bring up in specific geographical locations depending on their financial culture. Other than that, the paper looks good.”
We thank you very much for your comments on our article and appreciate your valuable advices. We would like to respond to your comments as follows:
Following your advice, we have added a section to discuss the economic and management implications of subsidies and some of the controversies encountered in international trade.
In the sixth part of the article, we make the following statement:
“Subsidies are relatively common in emerging market economies such as China, India, Brazil and Vietnam. In order to promote economic development, attract and accept industrial transfer from developed countries, these countries tolerated heavy polluting industries in the early stage. However, with the continuous development of the economies of these countries, the public's awareness of environmental protection has gradually increased. The government needs to balance economic interests and environmental impacts, eliminate or transfer heavy polluting industries, and take a series of measures to help and encourage some industries with less serious pollution and good economic benefits to improve their green level. Common subsidies include appropriation, tax allowance and exemption,priority and low interest loans. In addition, in international trade, export subsidies undoubtedly enhance the competitive advantage of their products and are vulnerable to countervailing measures by importing countries on the grounds of unfair competition. The green subsidies are classified as non-actionable subsidies by the agreement on Agreement on Subsidies and Countervailing Measures (SCM) implemented later, that is, allowed by WTO, and the member who suffered from the subsidies cannot appeal to DSB or implement corresponding countervailing measures. However, the complexity of international trade is beyond the reach of this agreement, and trade disputes caused by subsidies occur from time to time.
For developed countries, due to the mature market economic system, sound legal system of environmental protection and high threshold of market access, some industries with heavy pollution are excluded from the country. These countries mainly subsidize the production and use of renewable energy. Many countries offer very attractive subsidies to users who purchase new energy vehicles in order to promote the popularity of new energy vehicles. For example, the policy introduced in spring 2016 in Germany subsidizes 4,000 euros for users who purchase new electric vehicles and 3,000 euros for users who buy plug-in hybrid vehicles. The British subsidy policy is simpler and clearer. Regardless of the energy-driven mode of the vehicle, as long as the vehicle's electric endurance can exceed 70 miles, it can receive a 35% subsidy of the price, but the ceiling is 4,500 euros. If the vehicle's pure electric endurance is less than 70 miles, but above 10 miles, the subsidy ceiling will be reduced to 2500 euros. While other countries are offering incentives and exemptions based on energy-driven vehicles, France's incentives are based on carbon dioxide emissions from vehicles. Whether pure electric or hybrid vehicles, vehicles with CO2 emissions below 20g/km will receive a subsidy of 6000 euros. If the emission is higher than 21g/km and less than 60g/km, 1000 euros will be obtained. The "feed-in tariff" policy, which prevails in the EU, has greatly stimulated people's enthusiasm to participate in photovoltaic power generation and wind power generation. Up to now, nearly one third of energy consumption in the EU is provided by renewable energy sources. Accordingly, the EU has ambitiously set a target of 100% renewable energy supply for energy consumption by 2050. It is worth mentioning that the "feed-in tariff" scheme plans to gradually reduce until eliminate the subsidies for renewable energy supply, in order to encourage enterprises to carry out technological innovation to reduce production and operation costs. Similar to that, our future research will consider the dynamic subsidy situation.
The government's green subsidy plays an important role in promoting green investment and green development, but it also increases the government's financial burden. Some alternative schemes can also be tried, such as the new-energy vehicle credit program and corporate average fuel consumption regulation (dual-credit policy) implemented by the Chinese government to promote the development of the new energy automobile industry. Credit policy), or other indirect interventions, such as environmental taxes or the cap-and-trade policy, or the combination of these interventions.
Generally speaking, green subsidy is conducive to improving the ecological environment and achieving sustainable development of resources and ecological environment, which is worth advocating for both developed and developing countries. For developing countries, green subsidies also play a role in promoting the upgrading of industrial structure, balancing economic and environmental benefits. Essentially, green subsidy is a measure to resolve the external diseconomy of pollution emission and promote the internalization of social cost of enterprise pollution emission.”
Thank you again for your kind help.
Sincerely Yours

Reviewer 2 Report
Reading the title and having a first glance at the abstract it looked a promising paper, as it addresses a major supply chain management topic that has received extensive scientific interest lately, namely optimal GSC technology investment decision. The topic is highly relevant and fit well into the scope of the “Sustainability” Journal. The article is very interesting for the readers of the journal, but a number of improvements have to be made before published. In this light, I suggest a major revision of the article in order to merit publication. I hope that the criticisms I present below in bullet form will help the author improve the paper.
My main objections for publishing the paper in its current form are the following:
- Within the introductory section, the authors should provide clear and detailed analysis of the basic scope of the paper. Furthermore, the authors should elaborate more on Section1 regarding the problem statement.
- The authors should give a more detailed analysis and clarifications on the methodological advances of the presented approach. I am not completely convinced about the innovation of the methodological approach adopted in the proposed paper.
- Based on the above, the authors should provide a clearer structure of the presented study. To be more specific, the authors should separate the literature review and preliminaries. They should create a distinct section of methodological approach. Within this section the authors should provide a flowchart of the adopted methodology in order to illustrate a clear view to the readers. I would suggest the creation of a descriptive flowchart depicting the major steps of the proposed methodology. The exact flow of the methodology framework that have been utilized within the context of this study is not clear.
- I would expect more discussion within the justification of the overview of the results (Section 5 Numerical examples and discussions). It is not clear the outcome and the advantages of the experiments.
- In the conclusion’s section, I would expect some more managerial insights and general comments, rather than a repetition of study results. The authors should clearly reconstruct this section.
- As a final comment, I should repeat that the paper needs further explanations. I expect more discussion, more interpretation, based on the above comments. In principal the paper should convince that it does make methodological advances. In conclusion, even though the issue addressed by the manuscript is very important the methodology proposed is not convincing without addressing first the points highlighted above.
- Last but not least, linguistically, the paper needs significant improvements. It needs a proof reader to increase the quality of English.
Author Response
Dear Professor,
On behalf of my co-authors, we thank you very much for giving us an opportunity to revise our manuscript, we appreciate you very much for your positive and constructive comments and suggestions on our manuscript entitled “Closed-loop Supply Chain Network Equilibrium Model with Subsidy on GSC Technology Investment” (sustainability-554868).
We would like to respond to your comments as follows.
Within the introductory section, the authors should provide clear and detailed analysis of the basic scope of the paper. Furthermore, the authors should elaborate more on Section1 regarding the problem statement.
We focus our research topic on “the long-term utility of government subsidies to businesses.” After all, long-term subsidies and long-term investments are different from short-term ones. As the long-term investment mentioned in the introduction, it can save the production costs of enterprises and enable enterprises to obtain Economic benefits. So we added the following expression to the introduction:
“Technology investment and government subsidies have long-term effects. GSC investment may increase the cost and accounting risk of enterprises in the short-term, but in the long-term, it can save costs, increase profits and enhance the competitiveness of enterprises due to the expansion of demand for green products which driven by the improvement of consumers' environmental awareness. Government subsidies may not be effective in the short-term, but in the long-term, they may mobilize the enthusiasm of enterprises and affect the allocation of resources. How to depict the long-term effect of government subsidies for technology investment in GSC is a problem worth discussing, which is also the focus of this paper. In addition, market structure also has an important impact on government subsidies. Does the increase of market members make enterprises more willing to expand their investment scale to obtain more subsidies, or is investment weakened by the scarcity of resources and the limitation of demand? This is also one of the problems studied in this paper. Moreover, in the supply chain with many members, the strength of these enterprises is often different. This paper also studies the influence of government subsidies on the behavior and interests of different enterprises when the strength of enterprises is asymmetric.”
We sum up the problems to be solved as follows:
“How to resolve the dilemma encountered by SMEs in GSC technology investment, this paper has made some attempts and studies from the perspective of government subsidies, in order to ultimately achieve long-term stable and sustainable development of economic, social and ecological environment.”
The authors should give a more detailed analysis and clarifications on the methodological advances of the presented approach. I am not completely convinced about the innovation of the methodological approach adopted in the proposed paper.
In the part of literature review, we illustrate in detail that the advantage of "variational inequality" is that it can deal with the equilibrium problem of multi-decision-makers and multi-decision variables. It is very suitable for solving the multi-stage and multi-level network equilibrium problem in this paper. In addition, this research method has been very mature, there are a large number of literatures can be consulted. Therefore, we do not think it necessary to set up a separate section to introduce the origin and development of this research method.
Based on the above, the authors should provide a clearer structure of the presented study. To be more specific, the authors should separate the literature review and preliminaries. They should create a distinct section of methodological approach. Within this section the authors should provide a flowchart of the adopted methodology in order to illustrate a clear view to the readers. I would suggest the creation of a descriptive flowchart depicting the major steps of the proposed methodology. The exact flow of the methodology framework that have been utilized within the context of this study is not clear.
We listened to the your opinion and added a technical roadmap for the paper and explain it in detail.
I would expect more discussion within the justification of the overview of the results (Section 5 Numerical examples and discussions). It is not clear the outcome and the advantages of the experiments.
The conclusion responds to the problems mentioned in the introduction. Through the analysis of examples, we conclude that the long-term effect of government subsidies can effectively promote enterprises to invest in green supply chain technology. The more fierce competition brought by the increase of supply chain network members will weaken the effect of government subsidies, rather than make enterprises expand technology investment to earn more subsidies. Manufacturers with cost advantage will increase the investment in GSC technology, gain more profits, and improve the overall technology level of the supply chain. We think it's not easy to get a conclusion that violates common sense. It's also a good result to use the model to draw a conclusion that is consistent with common sense.
In the conclusion’s section, I would expect some more managerial insights and general comments, rather than a repetition of study results. The authors should clearly reconstruct this section.
Following your advice, we have added a section to discuss the economic and management implications of subsidies and some of the controversies encountered in international trade.
As a final comment, I should repeat that the paper needs further explanations. I expect more discussion, more interpretation, based on the above comments. In principal the paper should convince that it does make methodological advances. In conclusion, even though the issue addressed by the manuscript is very important the methodology proposed is not convincing without addressing first the points highlighted above.
We have followed your advice and made great efforts in these areas.
Last but not least, linguistically, the paper needs significant improvements. It needs a proof reader to increase the quality of English.
After all, English is not our native language, and the presentation of our paper may not be very authentic, but please understand. If this article is adopted, we will ask the editor for a fee-based polishing service.
Thank you again for your kind help.
Sincerely Yours

Round 2
Reviewer 2 Report
Despite the fact that the authors has modified the manuscript adequately based on the provided comments, the manuscript needs a native speaker in order to increase the quality of English language. Linguistically, the paper still needs significant improvements.
Author Response
Dear Professor,
We thank you very much for your patience and meticulousness and appreciate your serious work attitude. For the manuscript after the second round of revision, we make the following explanations:
We have commissioned MDPI's related services to revise the grammar and spelling of the article. In the revised browsing mode, the underlined red, green and blue text represents the added content, while the underlined above color text represents the deleted content. The 20th, 56th, and 145th lines of the revised browsing mode were modified to make the statement more in line with our intention. Fixed two spelling errors in Figure 2 and changed "Dicision variables" to "Decision variables"“. Fixed two spelling errors in Figure 3 and changed "Opttional object" to "Optimal objective". In the revised browsing mode, lines 626 to 632 give the meaning of the symbols in Table 2. Due to the failure of the reference editing software NoteExpress, the citation of the full text was entered manually, without setting the mode of cross-reference.
Thank you again for your kind help and look forward to your reply.
Sincerely Yours
